# Distributionally Robust Fair Principal Components via Geodesic Descents

**Hieu Vu, Toan Tran**
VinAI Research, Vietnam

**Man-Chung Yue**
Hong Kong Polytechnic University

**Viet Anh Nguyen**
VinAI Research, Vietnam

## Abstract

Principal component analysis is a simple yet useful dimensionality reduction technique in modern machine learning pipelines. In consequential domains such as college admission, healthcare and credit approval, it is imperative to take into account emerging criteria such as the fairness and the robustness of the learned projection. In this paper, we propose a distributionally robust optimization problem for principal component analysis which internalizes a fairness criterion in the objective function. The learned projection thus balances the trade-off between the total reconstruction error and the reconstruction error gap between subgroups, taken in the min-max sense over all distributions in a moment-based ambiguity set. The resulting optimization problem over the Stiefel manifold can be efficiently solved by a Riemannian subgradient descent algorithm with a sub-linear convergence rate. Our experimental results on real-world datasets show the merits of our proposed method over state-of-the-art baselines.

## 1 Introduction

Machine learning models are ubiquitous in our daily lives and supporting the decision-making process in diverse domains. With their flourishing applications, there also surface numerous concerns regarding the fairness of the models' outputs (Mehrabi et al., 2021). Indeed, these models are prone to biases due to various reasons (Barocas et al., 2018). First, the collected training data is likely to include some demographic disparities due to the bias in the data acquisition process (e.g., conducting surveys on a specific region instead of uniformly distributed places), or the imbalance of observed events at a specific period of time. Second, because machine learning methods only care about data statistics and are objective driven, groups that are under-represented in the data can be neglected in exchange for a better objective value. Finally, even human feedback to the predictive models can also be biased, e.g., click counts are human feedback to recommendation systems but they are highly correlated with the menu list suggested previously by a potentially biased system. Real-world examples of machine learning models that amplify biases and hence potentially cause unfairness are commonplace, ranging from recidivism prediction giving higher false positive rates for African-American[1] to facial recognition systems having large error rate for women[2].

To tackle the issue, various fairness criteria for supervised learning have been proposed in the literature, which encourage the (conditional) independence of the model's predictions on a particular sensitive attribute (Dwork et al., 2012; Hardt et al., 2016b; Kusner et al., 2017; Chouldechova, 2017; Verma & Rubin, 2018; Berk et al., 2021). Strategies to mitigate algorithmic bias are also investigated for all stages of the machine learning pipelines (Berk et al., 2021). For the pre-processing steps, (Kamiran & Calders, 2012) proposed reweighting or resampling techniques to achieve statistical parity between subgroups; in the training steps, fairness can be encouraged by adding constraints (Donini et al., 2018) or regularizing the original objective function (Kamishima et al., 2012; Zemel et al., 2013); and in the post-processing steps, adjusting classification threshold by examining black-box models over a holdout dataset can be used (Hardt et al., 2016b; Wei et al., 2019).

Since biases may already exist in the raw data, it is reasonable to demand machine learning pipelines to combat biases as early as possible. We focus in this paper on the Principal Component Analysis (PCA), which is a fundamental dimensionality reduction technique in the early stage of the

---

[1] https://www.propublica.org/article/machine-bias-risk-assessments-in-criminal-sentencing
[2] https://news.mit.edu/2018/study-finds-gender-skin-type-bias-artificial-intelligence-systems-0212

pipelines (Pearson, 1901; Hotelling, 1933). PCA finds a linear transformation that embeds the original data into a lower-dimensional subspace that maximizes the variance of the projected data. Thus, PCA may amplify biases if the data variability is different between the majority and the minority subgroups, see an example in Figure 1. A naive approach to promote fairness is to train one independent transformation for each subgroup. However, this requires knowing the sensitive attribute of each sample, which would raise disparity concerns. On the contrary, using a single transformation for all subgroups is "group-blinded" and faces no discrimination problem (Lipton et al., 2018).

Learning a fair PCA has attracted attention from many fields from machine learning, statistics to signal processing. Samadi et al. (2018) and Zalcberg & Wiesel (2021) propose to find the principal components that minimize the maximum subgroup reconstruction error; the min-max formulations can be relaxed and solved as semidefinite programs. Olfat & Aswani (2019) propose to learn a transformation that minimizes the possibility of predicting the sensitive attribute from the projected data. Apart from being a dimensionality reduction technique, PCA can also be thought of as a representation learning toolkit. Viewed in this way, we can also consider a more general family of fair representation learning methods that can be applied before any further analysis steps. There are a number of works develop towards this idea (Kamiran & Calders, 2012; Zemel et al., 2013; Calmon et al., 2017; Feldman et al., 2015; Beutel et al., 2017; Madras et al., 2018; Zhang et al., 2018; Tantipongpipat et al., 2019), which apply a multitude of fairness criteria.

In addition, we also focus on the robustness criteria for the linear transformation. Recently, it has been observed that machine learning models are susceptible to small perturbations of the data (Goodfellow et al., 2014; Madry et al., 2017; Carlini & Wagner, 2017). These observations have fuelled many defenses using adversarial training (Akhtar & Mian, 2018; Chakraborty et al., 2018) and distributionally robust optimization (Rahimian & Mehrotra, 2019; Kuhn et al., 2019; Blanchet et al., 2021).

**Contributions.** This paper blends the ideas from the field of fairness in artifical intelligence and distributionally robust optimization. Our contributions can be described as follows.

- We propose the fair principal components which balance between the total reconstruction error and the absolute gap of reconstruction error between subgroups. Moreover, we also add a layer of robustness to the principal components by considering a min-max formulation that hedges against all perturbations of the empirical distribution in a moment-based ambiguity set.

- We provide the reformulation of the distributionally robust fair PCA problem as a finite-dimensional optimization problem over the Stiefel manifold. We provide a Riemannian gradient descent algorithm and show that it has a sub-linear convergence rate.

Figure 1 illustrates the qualitative comparison between (fair) PCA methods and our proposed method on a 2-dimensional toy example. The majority group (blue dots) spreads on the horizontal axis, while the minority group (yellow triangles) spreads on the slanted vertical axis. The nominal PCA (red) captures the majority direction to minimize the total error, while the fair PCA of Samadi et al. (2018) returns the diagonal direction to minimize the maximum subgroup error. Our fair PCA can probe the full spectrum in between these two extremes by sweeping through our penalization parameters appropriately. If we do not penalize the error gap between subgroups, we recover the PCA method; if we penalize heavily, we recover the fair PCA of Samadi et al. (2018). Extensive numerical results on real datasets are provided in Section 5. Proofs are relegated to the appendix.

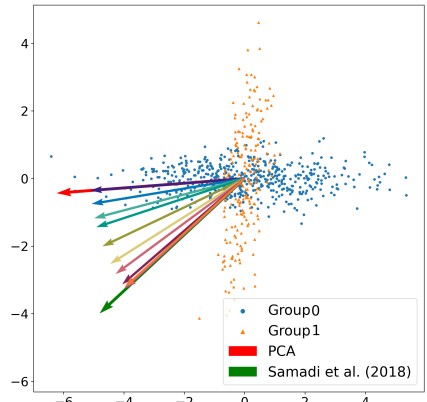

Figure 1: Nominal PCA (red arrow), fair PCA by Samadi et al. (2018) (green arrow), and our spectrum of fair PCA (shorter arrows). Arrows show directions and are not normalized to unit length.

## 2   FAIR PRINCIPAL COMPONENT ANALYSIS

### 2.1   PRINCIPAL COMPONENT ANALYSIS

We first briefly revisit the classical PCA. Suppose that we are given a collection of $N$ i.i.d. samples $\{\hat{x}_i\}_{i=1}^N$ generated by some underlying distribution $\mathbb{P}$. For simplicity, we assume that both the empirical and population mean are zero vectors. The goal of PCA is to find a $k$-dimensional linear subspace of $\mathbb{R}^d$ that explains as much variance contained in the data $\{\hat{x}_i\}_{i=1}^N$ as possible, where $k < d$ is a given integer. More precisely, we parametrize $k$-dimensional linear subspaces by orthonormal matrices, *i.e.*, matrices whose columns are orthogonal and have unit Euclidean norm. Given any such matrix $V$, the associated $k$-dimensional subspace is the one spanned by the columns of $V$. The projection matrix onto the subspace is $VV^\top$, and hence the variance of the projected data is given by $\mathrm{tr}\left(VV^\top \Xi\Xi^\top\right)$, where $\Xi = [\hat{x}_1, \cdots, \hat{x}_N] \in \mathbb{R}^{d \times N}$ is the data matrix. By a slight abuse of terminology, sometimes we refer to $V$ as the projection matrix. The problem of PCA then reads

$$\max_{V \in \mathbb{R}^{d \times k}, V^\top V = I_k} \mathrm{tr}\left(VV^\top \Xi\Xi^\top\right). \tag{1}$$

For any vector $X \in \mathbb{R}^d$ and orthonormal matrix $V$, denote by $\ell(V, X)$ the reconstruction error, *i.e.*,

$$\ell(V, X) = \|X - VV^\top X\|_2^2 = X^\top (I_d - VV^\top)X.$$

The problem of PCA can alternatively be formulated as a stochastic optimization problem

$$\min_{V \in \mathbb{R}^{d \times k}, V^\top V = I_k} \mathbb{E}_{\hat{\mathbb{P}}}[\ell(V, X)], \tag{2}$$

where $\hat{\mathbb{P}}$ is the empirical distribution associated with the samples $\{\hat{x}_i\}_{i=1}^N$ and $X \sim \hat{\mathbb{P}}$. It is well-known that PCA admits an analytical solution. In particular, the optimal solution to problem (2) (and also problem (1)) is given by any orthonormal matrix whose columns are the eigenvectors associated with the $k$ largest eigenvalues of the sample covariance matrix $\Xi\Xi^\top$.

### 2.2   FAIR PRINCIPAL COMPONENT ANALYSIS

In the fair PCA setting, we are also given a discrete sensitive attribute $A \in \mathcal{A}$, where $A$ may represent features such as race, gender or education. We consider binary attribute $A$ and let $\mathcal{A} = \{0, 1\}$. A straightforward idea to define fairness is to require the (strict) balance of a certain objective between the two groups. For example, this is the strategy in Hardt et al. (2016a) for developing fair supervised learning algorithms. A natural objective to balance in the PCA context is the reconstruction error.

**Definition 2.1** (Fair projection). Let $\mathbb{Q}$ be an arbitrary distribution of $(X, A)$. A projection matrix $V \in \mathbb{R}^{d \times k}$ is fair relative to $\mathbb{Q}$ if the conditional expected reconstruction error is equal between subgroups, *i.e.*, $\mathbb{E}_{\mathbb{Q}}[\ell(V, X)|A = a] = \mathbb{E}_{\mathbb{Q}}[\ell(V, X)|A = a']$ for any $(a, a') \in \mathcal{A} \times \mathcal{A}$.

Unfortunately, Definition 2.1 is too stringent: for a general probability distribution $\mathbb{Q}$, it is possible that there exists no fair projection matrix $V$.

**Proposition 2.2** (Impossibility result). For any distribution $\mathbb{Q}$ on $\mathcal{X} \times \mathcal{A}$, there exists a fair projection matrix $V \in \mathbb{R}^{d \times k}$ relative to $\mathbb{Q}$ if and only if $\mathrm{rank}(\mathbb{E}_{\mathbb{Q}}[XX^\top|A = 0] - \mathbb{E}_{\mathbb{Q}}[XX^\top|A = 1]) \leq k$.

One way to circumvent the impossibility result is to relax the requirement of strict balance to approximate balance. In other words, an inequality constraint of the following form is imposed:

$$|\mathbb{E}_{\mathbb{Q}}[\ell(V, X)|A = a] - \mathbb{E}_{\mathbb{Q}}[\ell(V, X)|A = a']| \leq \epsilon \qquad \forall (a, a') \in \mathcal{A} \times \mathcal{A},$$

where $\epsilon > 0$ is some prescribed fairness threshold. This approach has been adopted in other fair machine learning settings, see Donini et al. (2018) and Agarwal et al. (2019) for example.

In this paper, instead of imposing the fairness requirement as a constraint, we penalize the unfairness in the objective function. Specifically, for any projection matrix $V$, we define the unfairness as the absolute difference between the conditional loss between two subgroups:

$$\mathbb{U}(V, \mathbb{Q}) \triangleq |\mathbb{E}_{\mathbb{Q}}[\ell(V, X)|A = 0] - \mathbb{E}_{\mathbb{Q}}[\ell(V, X)|A = 1]|.$$

We thus consider the following fairness-aware PCA problem

$$\min_{V \in \mathbb{R}^{d \times k}, V^\top V = I_k} \mathbb{E}_{\hat{\mathbb{P}}}[\ell(V, X)] + \lambda \mathbb{U}(V, \hat{\mathbb{P}}), \tag{3}$$

where $\lambda \geq 0$ is a penalty parameter to encourage fairness. Note that for fair PCA, the dataset is $\{(\hat{x}_i, \hat{a}_i)\}_{i=1}^N$ and hence the empirical distribution $\hat{\mathbb{P}}$ is given by $\hat{\mathbb{P}} = \frac{1}{N} \sum_{i=1}^N \delta_{(\hat{x}_i, \hat{a}_i)}$.

## 3 DISTRIBUTIONALLY ROBUST FAIR PCA

The weakness of empirical distribution-based stochastic optimization has been well-documented, see (Smith & Winkler, 2006; Homem-de Mello & Bayraksan, 2014). In particular, due to overfitting, the out-of-sample performance of the decision, prediction, or estimation obtained from such a stochastic optimization model is unsatisfactory, especially in the low sample size regime. Ideally, we could improve the performance by using the underlying distribution $\mathbb{P}$ instead of the empirical distribution $\hat{\mathbb{P}}$. But the underlying distribution $\mathbb{P}$ is unavailable in most practical situations, if not all. Distributional robustification is an emerging approach to handle this issue and has been shown to deliver promising out-of-sample performance in many applications (Delage & Ye, 2010; Namkoong & Duchi, 2017; Kuhn et al., 2019; Rahimian & Mehrotra, 2019). Motivated by the success of distributional robustification, especially in machine learning Nguyen et al. (2019); Taskesen et al. (2021), we propose a robustified version of model (3), called the distributionally robust fairness-aware PCA:

$$\min_{V \in \mathbb{R}^{d \times k}, V^\top V = I_k} \sup_{\mathbb{Q} \in \mathbb{B}(\hat{\mathbb{P}})} \mathbb{E}_{\mathbb{Q}}[\ell(V, X)] + \lambda \mathbb{U}(V, \mathbb{Q}), \tag{4}$$

where $\mathbb{B}(\hat{\mathbb{P}})$ is a set of probability distributions similar to the empirical distribution $\hat{\mathbb{P}}$ in a certain sense, called the ambiguity set. The empirical distribution $\hat{\mathbb{P}}$ is also called the nominal distribution. Many different ambiguity sets have been developed and studied in the optimization literature, see Rahimian & Mehrotra (2019) for an extensive overview.

### 3.1 THE WASSERSTEIN-TYPE AMBIGUITY SET

To present our ambiguity set and main results, we need to introduce some definitions and notations.

**Definition 3.1** (Wasserstein-type divergence). The divergence $\mathbb{W}$ between two probability distributions $\mathbb{Q}_1 \sim (\mu_1, \Sigma_1) \in \mathbb{R}^d \times \mathbb{S}_+^d$ and $\mathbb{Q}_2 \sim (\mu_2, \Sigma_2) \in \mathbb{R}^d \times \mathbb{S}_+^d$ is defined as

$$\mathbb{W}(\mathbb{Q}_1 \parallel \mathbb{Q}_2) \triangleq \|\mu_1 - \mu_2\|_2^2 + \text{tr}\left(\Sigma_1 + \Sigma_2 - 2\left(\Sigma_2^{\frac{1}{2}} \Sigma_1 \Sigma_2^{\frac{1}{2}}\right)^{\frac{1}{2}}\right).$$

The divergence $\mathbb{W}$ coincides with the *squared* type-2 Wasserstein distance between two Gaussian distributions $\mathcal{N}(\mu_1, \Sigma_1)$ and $\mathcal{N}(\mu_2, \Sigma_2)$ (Givens & Shortt, 1984). One can readily show that $\mathbb{W}$ is non-negative, and it vanishes if and only if $(\mu_1, \Sigma_1) = (\mu_2, \Sigma_2)$, which implies that $\mathbb{Q}_1$ and $\mathbb{Q}_2$ have the same first- and second-moments. Recently, distributional robustification with Wasserstein-type ambiguity sets has been applied widely to various problems including domain adaption (Taskesen et al., 2021), risk measurement (Nguyen et al., 2021b) and statistical estimation (Nguyen et al., 2021a). The Wasserstein-type divergence in Definition 3.1 is also related to the theory of optimal transport with its applications in robust decision making (Mohajerin Esfahani & Kuhn, 2018; Blanchet & Murthy, 2019; Yue et al., 2021) and potential applications in fair machine learning (Taskesen et al., 2020; Si et al., 2021; Wang et al., 2021).

Recall that the nominal distribution is $\hat{\mathbb{P}} = \frac{1}{N} \sum_{i=1}^{N} \delta_{(\hat{x}_i, \hat{a}_i)}$. For any $a \in \mathcal{A}$, its conditional distribution given $A = a$ is given by

$$\hat{\mathbb{P}}_a = \frac{1}{|\mathcal{I}_a|} \sum_{i \in \mathcal{I}_a} \delta_{x_i}, \quad \text{where} \quad \mathcal{I}_a \triangleq \{i \in \{1, \dots, N\} : a_i = a\}.$$

We also use $(\hat{\mu}_a, \hat{\Sigma}_a)$ to denote the empirical mean vector and covariance matrix of $X$ given $A = a$:

$$\hat{\mu}_a = \mathbb{E}_{\hat{\mathbb{P}}_a}[X] = \mathbb{E}_{\hat{\mathbb{P}}}[X|A = a] \quad \text{and} \quad \hat{\Sigma}_a + \hat{\mu}_a \hat{\mu}_a^\top = \mathbb{E}_{\hat{\mathbb{P}}_a}[XX^\top] = \mathbb{E}_{\hat{\mathbb{P}}}[XX^\top|A = a].$$

For any $a \in \mathcal{A}$, the empirical marginal distribution of $A$ is denoted by $\hat{p}_a = |\mathcal{I}_a|/N$.

Finally, for any set $\mathcal{S}$, we use $\mathcal{P}(\mathcal{S})$ to denote the set of all probability distributions supported on $\mathcal{S}$. For any integer $k$, the $k$-by-$k$ identity matrix is denoted $I_k$. We then define our ambiguity set as

$$\mathbb{B}(\hat{\mathbb{P}}) \triangleq \left\{ \mathbb{Q} \in \mathcal{P}(\mathcal{X} \times \mathcal{A}) : \begin{array}{l} \exists \mathbb{Q}_a \in \mathcal{P}(\mathcal{X}) \text{ such that:} \\ \mathbb{Q}(\mathbb{X} \times \{a\}) = \hat{p}_a \mathbb{Q}_a(\mathbb{X}) \quad \forall \mathbb{X} \subseteq \mathbb{R}^d, \ a \in \mathcal{A} \\ \mathbb{W}(\mathbb{Q}_a, \hat{\mathbb{P}}_a) \leq \varepsilon_a \quad \forall a \in \mathcal{A} \end{array} \right\}, \tag{5}$$

where $\mathbb{Q}_a$ is the conditional distribution of $X|A = a$. Intuitively, each $\mathbb{Q} \in \mathbb{B}(\hat{\mathbb{P}})$ is a joint distribution of the random vector $(X, A)$, formed by taking a mixture of conditional distributions $\mathbb{Q}_a$ with mixture weight $\hat{p}_a$. Each conditional distribution $\mathbb{Q}_a$ is constrained in an $\varepsilon_a$-neighborhood of the nominal conditional distribution $\hat{\mathbb{P}}_a$ with respect to the $\mathbb{W}$ divergence. Because the loss function $\ell$ is a quadratic function of $X$, the (conditional) expected losses only involve the first two moments of $X$, and thus prescribing the ambiguity set using $\mathbb{W}$ would suffice for the purpose of robustification.

## 3.2 REFORMULATION

We now present the reformulation of problem (4) under the ambiguity set $\mathbb{B}(\hat{\mathbb{P}})$.

**Theorem 3.2** (Reformulation). Suppose that for any $a \in \mathcal{A}$, either of the following two conditions holds:

(i) Marginal probability bounds: $0 \leq \lambda \leq \hat{p}_a$,

(ii) Eigenvalue bounds: the empirical second moment matrix $\hat{M}_a = \frac{1}{N_a} \sum_{i \in \mathcal{I}_a} \hat{x}_i \hat{x}_i^\top$ satisfies $\sum_{j=1}^{d-k} \sigma_j(\hat{M}_a) \geq \varepsilon_a$, where $\sigma_j(\hat{M}_a)$ is the $j$-th smallest eigenvalues of $\hat{M}_a$.

Then problem (4) is equivalent to

$$\min_{V \in \mathbb{R}^{d \times k}, V^\top V = I_k} \max\{J_0(V), J_1(V)\}, \tag{6a}$$

where for each $(a, a') \in \{(0, 1), (1, 0)\}$, the function $J_a$ is defined as

$$J_a(V) = \kappa_a + \theta_a \sqrt{\langle I_d - VV^\top, \hat{M}_a \rangle} + \vartheta_{a'} \sqrt{\langle I_d - VV^\top, \hat{M}_{a'} \rangle} + \langle I_d - VV^\top, C_a \rangle, \tag{6b}$$

and the parameters $\kappa \in \mathbb{R}$, $\theta \in \mathbb{R}$, $\vartheta \in \mathbb{R}$ and $C \in \mathbb{S}_+^d$ are defined as

$$\begin{aligned} \kappa_a &= (\hat{p}_a + \lambda)\varepsilon_a + (\hat{p}_{a'} - \lambda)\varepsilon_{a'}, \quad \theta_a = 2|\hat{p}_a + \lambda|\sqrt{\varepsilon_a}, \quad \vartheta_{a'} = 2|\hat{p}_{a'} - \lambda|\sqrt{\varepsilon_{a'}}, \\ C_a &= (\hat{p}_a + \lambda)\hat{M}_a + (\hat{p}_{a'} - \lambda)\hat{M}_{a'}. \end{aligned} \tag{6c}$$

We now briefly explain the steps that lead to the results in Theorem 3.2. Letting

$$J_0(V) = \sup_{\mathbb{Q} \in \mathbb{B}(\hat{\mathbb{P}})} (\hat{p}_0 + \lambda)\mathbb{E}_\mathbb{Q}[\ell(V, X)|A = 0] + (\hat{p}_1 - \lambda)\mathbb{E}_\mathbb{Q}[\ell(V, X)|A = 1],$$

$$J_1(V) = \sup_{\mathbb{Q} \in \mathbb{B}(\hat{\mathbb{P}})} (\hat{p}_0 - \lambda)\mathbb{E}_\mathbb{Q}[\ell(V, X)|A = 0] + (\hat{p}_1 + \lambda)\mathbb{E}_\mathbb{Q}[\ell(V, X)|A = 1],$$

then by expanding the term $\mathbb{U}(V, \mathbb{Q})$ using its definition, problem (4) becomes

$$\min_{V \in \mathbb{R}^{d \times k}, V^\top V = I_k} \max\{J_0(V), J_1(V)\}.$$

Leveraging the definition the ambiguity set $\mathbb{B}(\hat{\mathbb{P}})$, for any pair $(a, a') \in \{(0, 1), (1, 0)\}$, we can decompose $J_a$ into two separate supremum problems as follows

$$J_a(V) = \sup_{\mathbb{Q}_a : \mathbb{W}(\mathbb{Q}_a, \hat{\mathbb{P}}_a) \leq \varepsilon_a} (\hat{p}_a + \lambda)\mathbb{E}_{\mathbb{Q}_a}[\ell(V, X)] + \sup_{\mathbb{Q}_{a'} : \mathbb{W}(\mathbb{Q}_{a'}, \hat{\mathbb{P}}_{a'}) \leq \varepsilon_{a'}} (\hat{p}_{a'} - \lambda)\mathbb{E}_{\mathbb{Q}_{a'}}[\ell(V, X)].$$

The next proposition asserts that each individual supremum in the above expression admits an analytical expression.

**Proposition 3.3** (Reformulation). Fix $a \in \mathcal{A}$. For any $\upsilon \in \mathbb{R}$, $\varepsilon_a \in \mathbb{R}_+$, it holds that

$$\sup_{\mathbb{Q}_a : \mathbb{W}(\mathbb{Q}_a, \hat{\mathbb{P}}_a) \leq \varepsilon_a} \upsilon \mathbb{E}_{\mathbb{Q}_a}[\ell(V, X)]$$

$$= \begin{cases} \upsilon \left( \sqrt{\langle I_d - VV^\top, \hat{M}_a \rangle} + \sqrt{\varepsilon_a} \right)^2 & \text{if } \upsilon \geq 0, \\ \upsilon \left( \sqrt{\langle I_d - VV^\top, \hat{M}_a \rangle} - \sqrt{\varepsilon_a} \right)^2 & \text{if } \upsilon < 0 \text{ and } \langle I_d - VV^\top, \hat{M}_a \rangle \geq \varepsilon_a, \\ 0 & \text{if } \upsilon < 0 \text{ and } \langle I_d - VV^\top, \hat{M}_a \rangle < \varepsilon_a. \end{cases}$$

The proof of Theorem 3.2 now follows by applying Proposition 3.3 to each term in $J_a$, and balance the parameters to obtain (6c). A detailed proof is relegated to the appendix. In the next section, we study an efficient algorithm to solve problem (6a).

**Remark 3.4** (Recovery of the nominal PCA). If $\lambda = 0$ and $\varepsilon_a = 0 \ \forall a \in \mathcal{A}$, our formulation (4) becomes the standard PCA problem (2). In this case, our robust fair principal components reduce to the standard principal components. On the contrary, existing fair PCA methods such as Samadi et al. (2018) and Olfat & Aswani (2019) cannot recover the standard principal components.

## 4 RIEMANNIAN GRADIENT DESCENT ALGORITHM

The distributionally robust fairness-aware PCA problem (4) is originally an infinite-dimensional min-max problem. Indeed, the inner maximization problem in (4) optimizes over the space of probability measures. Thanks to Theorem 3.2, it is reduced to the simpler finite-dimensional minimax problem (6a), where the inner problem is only a maximization over two points. Problem (6a) is, however, still challenging as it is a non-convex optimization problem over a non-convex feasible region defined by the orthogonality constraint $V^\top V = I_d$. The purpose of this section is to devise an efficient algorithm for solving problem (6a) to local optimality based on Riemannian optimization.

### 4.1 REPARAMETRIZATION

As mentioned above, the non-convexity of problem (6a) comes from both the objective function and the feasible region. It turns out that we can get rid of the non-convexity of the objective function via a simple change of variables. To see that, we let $U \in \mathbb{R}^{d \times (d-k)}$ be an orthonormal matrix complement to $V$, that is, $U$ and $V$ satisfy $UU^\top + VV^\top = I_d$. Thus, we can express the objective function $J$ via

$$J(V) = F(U) \triangleq \max\{F_0(U), F_1(U)\},$$

where for $(a, a') \in \{(0, 1), (1, 0)\}$, the function $F_a$ is defined as

$$F_a(U) \triangleq \kappa_a + \theta_a \sqrt{\langle UU^\top, \hat{M}_a \rangle} + \vartheta_{a'} \sqrt{\langle UU^\top, \hat{M}_{a'} \rangle} + \langle UU^\top, C_a \rangle.$$

Moreover, letting $\mathcal{M} \triangleq \{U \in \mathbb{R}^{d \times (d-k)} : U^\top U = I_{d-k}\}$, we can re-express problem (6a) as

$$\min_{U \in \mathcal{M}} F(U). \tag{7}$$

The set $\mathcal{M}$ of problem (7) is a Riemannian manifold, called the Stiefel manifold (Absil et al., 2007, Section 3.3.2). It is then natural to solve (7) using a Riemannian optimization algorithms (Absil et al., 2007). In fact, problem (6a) itself (before the change of variables) can also be cast as a Riemannian optimization problem over another Stiefel manifold. The change of variables above might seem unnecessary. Nonetheless, the upshot of problem (7) is that the objective function $F$ is convex (in the traditional sense). This faciliates the application of the theoretical and algorithmic framework developed in Li et al. (2021) for (weakly) convex optimization over the Stiefel manifolds.

### 4.2 THE RIEMANNIAN SUBGRADIENT

Note that the objective function $F$ is non-smooth since it is defined as the maximum of two functions $F_0$ and $F_1$. To apply the framework in Li et al. (2021), we need to compute the Riemannian subgradient of the objective function $F$. Since the Stiefel manifold $\mathcal{M}$ is an embedded manifold in Euclidean space, the Riemannian subgradient of $F$ at any point $U \in \mathcal{M}$ is given by the orthogonal projection of the usual Euclidean subgradient onto the tangent space of the manifold $\mathcal{M}$ at the point $U$, see Absil et al. (2007, Section 3.6.1) for example.

**Lemma 4.1.** For any point $U \in \mathcal{M}$, let[3] $a_U \in \arg\max_{a \in \{0,1\}} F_a(U)$ and $a'_U = 1 - a_U$. Then, a Riemannian subgradient of the objective function $F$ at the point $U$ is given by

$$\mathrm{grad}F(U) = (I_d - UU^\top)\left(\frac{\theta_{a_U}}{\sqrt{\langle UU^\top, \hat{M}_{a_U} \rangle}}\hat{M}_{a_U}U + \frac{\vartheta_{a'_U}}{\sqrt{\langle UU^\top, \hat{M}_{a'_U} \rangle}}\hat{M}_{a'_U}U + 2C_{a_U}U\right).$$

---

[3] It is possible that the maximizer is not unique. In that case, choosing $a_U$ to be either 0 or 1 would work.

### 4.3 RETRACTIONS

Another important instrument required by the framework in Li et al. (2021) is a retraction of the Stiefel manifold $\mathcal{M}$. At each iteration, the point $U - \gamma\Delta$ obtained by moving from the current iterate $U$ in the opposite direction of the Riemannian gradient $\Delta$ may not lie on the manifold in general, where $\gamma > 0$ is the stepsize. In Riemannian optimization, this is circumvented by the concept of retraction. Given a point $U \in \mathcal{M}$ on the manifold, the Riemannian gradient $\Delta \in T_U\mathcal{M}$ (which must lie in the tangent space $T_U\mathcal{M}$) and a stepsize $\gamma$, the retraction map Rtr defines a point $\text{Rtr}_U(-\gamma\Delta)$ which is guaranteed to lie on the manifold $\mathcal{M}$. Roughly speaking, the retraction $\text{Rtr}_U(\cdot)$ approximates the geodesic curve through $U$ along the input tangential direction. For a formal definition of retractions, we refer the readers to (Absil et al., 2007, Section 4.1). In this paper, we focus on the following two commonly used retractions for Stiefel manifolds. The first one is the QR decomposition-based retraction using the Q-factor qf($\cdot$) in the QR decomposition:

$$\text{Rtr}_U^{\text{qf}}(\Delta) = \text{qf}(U + \Delta), \quad U \in \mathcal{M}, \Delta \in T_U\mathcal{M}.$$

The second one is the polar decomposition-based retraction

$$\text{Rtr}_U^{\text{polar}}(\Delta) = (U + \Delta)(I_{d-k} + \Delta^\top\Delta)^{-\frac{1}{2}}, \quad U \in \mathcal{M}, \Delta \in T_U\mathcal{M}. \tag{8}$$

### 4.4 ALGORITHM AND CONVERGENCE GUARANTEES

Associated with any choice of retraction Rtr is a concrete instantiation of the Riemannian subgradient descent algorithm for our problem (7), which is presented in Algorithm 1 with specific choice of the stepsizes $\gamma_t$ motivated by the theoretical results of (Li et al., 2021).

---

**Algorithm 1** Riemannian Subgradient Descent for (7)

---

1: **Input:** An initial point $U_0$, a number of iterations $\tau$ and a retraction Rtr : $(U, \Delta) \mapsto \text{Rtr}_U(\Delta)$.
2: **for** $t = 0, 1, \dots, \tau - 1$, **do**
3:      Find $a_t \triangleq \arg\max_{a \in \{0,1\}}\{F_a(U_t)\}$.
4:      Compute the Riemannian subgradient $\Delta_t = \text{grad}F(U_t)$ using the formula

$$\Delta_t = (I - U_t U_t^\top)\left(\frac{\theta_{a_t}}{\sqrt{\langle U_t U_t^\top, \hat{M}_{a_t}\rangle}}\hat{M}_{a_t}U_t + \frac{\vartheta_{a_t'}}{\sqrt{\langle U_t U_t^\top, \hat{M}_{a_t'}\rangle}}\hat{M}_{a_t'}U_t + 2C_{a_t}U_t\right).$$

5:      Set $U_{t+1} = \text{Rtr}_{U_t}(-\gamma_t\Delta_t)$, where the step-size $\gamma_t \equiv \frac{1}{\sqrt{\tau+1}}$ is constant.
6: **end for**
7: **Output:** $U_\tau$.

---

We now study the convergence guarantee of Algorithm 1. The following lemma shows that the objective function $F$ is Lipschitz continuous (with respect to the Riemannian metric on the Stiefel manifold $\mathcal{M}$) with an explicit Lipschitz constant $L$.

**Lemma 4.2** (Lipschitz continuity). *The function $F$ is $L$-Lipschitz continuous on $\mathcal{M}$, where $L > 0$ is given by*

$$L \triangleq \max\left\{\theta_0 \frac{\sigma_{\max}(\hat{M}_0)}{\sqrt{\sigma_{\min}(\hat{M}_0)}}, \theta_1 \frac{\sigma_{\max}(\hat{M}_1)}{\sqrt{\sigma_{\min}(\hat{M}_1)}}, \vartheta_0 \frac{\sigma_{\max}(\hat{M}_0)}{\sqrt{\sigma_{\min}(\hat{M}_0)}}, \vartheta_1 \frac{\sigma_{\max}(\hat{M}_1)}{\sqrt{\sigma_{\min}(\hat{M}_1)}},\right.$$
$$\left. 2\sqrt{d-k}\sigma_{\max}(C_0), 2\sqrt{d-k}\sigma_{\max}(C_1)\right\}. \tag{9}$$

We now proceed to show that Algorithm 1 enjoys a sub-linear convergence rate. To state the result, we define the Moreau envelope

$$F_\mu(U) \triangleq \min_{U' \in \mathcal{M}}\left\{F(U') + \frac{1}{2\mu}\|U' - U\|_F^2\right\},$$

where $\| \cdot \|_F$ denotes the Frobenius norm of a matrix. Also, to measure the progress of the algorithm, we need to introduce the proximal mapping on the Stiefel manifold (Li et al., 2021):

$$\mathrm{prox}_{\mu F}(U) \in \arg\min_{U' \in \mathcal{M}} \left\{ F(U') + \frac{1}{2\mu} \|U' - U\|_F^2 \right\}.$$

From Li et al. (2021, Equation (22)), we have that

$$\|\mathrm{grad}F(U)\|_F \leq \frac{\left\|\mathrm{prox}_{\mu F}(U) - U\right\|_F}{\mu} \triangleq \mathrm{gap}_\mu(U).$$

Therefore, the number $\mathrm{gap}_\mu(U)$ is a good candidate to quantify the progress of optimization algorithms for solving problem (7).

**Theorem 4.3** (Convergence guarantee). Let $\{U_t\}_{t=1,\ldots,\tau}$ be the sequence of iterates generated by Algorithm 1. Suppose that $\mu = 1/4L$, where $L$ is the Lipschitz constant of $F$ in (9). Then, we have

$$\min_{t=0,\ldots,\tau} \mathrm{gap}_\mu(U_t) \leq \frac{2\sqrt{F_\mu(U_0) - \min_U F_\mu(U) + 2L^3(L+1)}}{(\tau+1)^{1/4}}.$$

## 5 NUMERICAL EXPERIMENTS

We compare our proposed method, denoted `RFPCA`, against two state-of-the-art methods for fair PCA: 1) `FairPCA` Samadi et al. (2018)[4], and 2) `CFPCA` Olfat & Aswani (2019)[5] with both cases: only mean constraint, and both mean and covariance constraints. We consider a wide variety of datasets with ranging sample sizes and number of features. Further details about the datatasets can be found in Appendix C. The code for all experiments is available in supplementary materials. We include here some details about the hyper-parameters that we search in the cross-validation steps.

- `RFPCA`. We notice that the neighborhood size $\varepsilon_a$ should be inversely proportional to the size of subgroup $a$. Indeed, a subgroup with large sample size is likely to have more reliable estimate of the moment information. Then we parameterize the neighborhood size $\varepsilon_a$ by a common scalar $\alpha$, and we have $\varepsilon_a = \alpha/\sqrt{N_a}$, where $N_a$ is the number of samples in group $a$. We search $\alpha \in \{0.05, 0.1, 0.15\}$ and $\lambda \in \{0., 0.5, 1., 1.5, 2.0, 2.5\}$. For better convergence quality, we set the number of iteration for our subgradient descent algorithm to $\tau = 1000$ and also repeat the Riemannian descent for 20 randomly generated initial point $U_0$.

- `FairPCA`. According to Samadi et al. (2018), we only need tens of iterations for the multiplicative weight algorithm to provide good-quality solution; however, to ensure a fair comparison, we set the number of iterations to 1000 for the convergence guarantee. We search the learning rate $\eta$ of the algorithm from set of 17 values evenly spaced in $[0.25, 4.25]$ and $\{0.1\}$.

- `CFPCA`. Following Olfat & Aswani (2019), for the mean-constrained version of `CFPCA`, we search $\delta$ from $\{0., 0.1, 0.3, 0.5, 0.6, 0.7, 0.8, 0.9\}$, and for both the mean and covariance constrained version, we fix $\delta = 0$ while searching $\mu$ in $\{0.0001, 0.001, 0.01, 0.05, 0.5\}$.

**Trade-offs.** First, we examine the trade-off between the total reconstruction error and the gap between the subgroup error. In this experiment, we only compare our model with `FairPCA` and `CFPCA` mean-constraint version. We plot a pareto curve for each of them over the two criteria with different hyper-parameters (hyper-parameters test range are mentioned above). The whole datasets are used for training and evaluation. The results averaged over 5 runs are shown in Figure 2.

In testing methods with different principal components, we first split each dataset into training set and test set with equal size (50% each), the projection matrix of each method is learned from training set and tested over both sets. In this case, we only compare our method with traditional PCA and `FairPCA` method. We fix one set hyper-parameters for each method. For `FairPCA`, we set $\eta = 0.1$ and for `RFPCA` we set $\alpha = 0.15, \lambda = 0.5$, others hyper-parameters are kept as discussed before. The results are averaged over 5 different splits. Figure 3 shows the consistence of our method performing fair projections over different values of $k$. Our method (cross) exhibits smaller gap of subgroup errors. More results and discussions on the effect of $\varepsilon$ can be found in Appendix D.2.

---

[4] https://github.com/samirasamadi/Fair-PCA   [5] https://github.com/molfat66/FairML

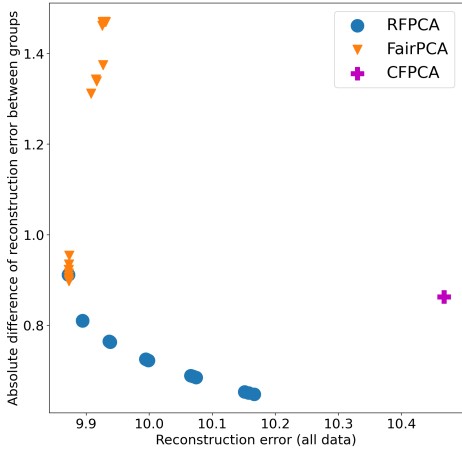

Figure 2: Pareto curves on Default Credit dataset (all data) with 3 principal components

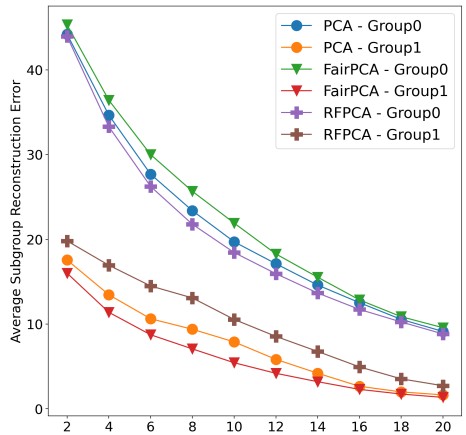

Figure 3: Subgroup average error with different $k$ on Biodeg dataset (Out-of-sample).

**Cross-validations.** Next, we report the performance of all methods based on three criteria: absolute difference between average reconstruction error between groups (ABDiff.), average reconstruction error of all data (ARE.), and the fairness criterion defined by Olfat & Aswani (2019) with respect to a linear SVM's classifier family ($\triangle\mathcal{F}_{Lin}$).[6] Due to the space constraint, we only include the first two criteria in the main text, see Appendix 4 for full results. To emphasize the generalization capacity of each algorithm, we split each dataset into a training set and a test set with ratio of $30\% - 70\%$ respectively, and only extract top three principal components from the training set. We find the best hyper-parameters by 3-fold cross validation, and prioritize the one giving minimum value of the summation (ABDiff.+ARE.). The results are averaged over 10 different training-testing splits. We report the performance on both training set (In-sample data) and test set (Out-of-sample data). The details results for Out-of-sample data is given in Table 1, more details about settings and performance can be found at Appendix D.

*Results.* Our proposed RFPCA method outperforms on 11 out of 15 datasets in terms of the subgroup error gap ABDiff, and 9 out of 15 with the totall error ARE. criterion. There are 5 datasets that RFPCA gives the best results for both criteria, and for the remaining datasets, RFPCA has small performance gaps compared with the best method.

Table 1: Out-of-sample errors on real datasets. Bold indicates the lowest error for each dataset.

| Dataset | RFPCA ABDiff. | RFPCA ARE. | FairPCA ABDiff. | FairPCA ARE. | CFPCA-Mean Con. ABDiff. | CFPCA-Mean Con. ARE. | CFPCA - Both Con. ABDiff. | CFPCA - Both Con. ARE. |
|---|---|---|---|---|---|---|---|---|
| Default Credit | 0.9483 | **10.3995** | 1.4401 | 10.4439 | **0.9367** | 10.9451 | 3.3359 | 22.0310 |
| Biodeg | **23.0066** | **33.8571** | 27.5159 | 34.6184 | 29.1728 | 37.6052 | 37.9533 | 50.7090 |
| E. Coli | 1.1500 | **1.7210** | 1.5280 | 2.4799 | **1.1005** | 2.9466 | 5.1275 | 5.6674 |
| Energy | **0.0125** | 0.2238 | 0.0138 | **0.2225** | 0.1229 | 2.7318 | 0.1001 | 7.9511 |
| German Credit | 2.0588 | **43.9032** | **1.3670** | 44.0064 | 1.7845 | 43.9648 | 1.4955 | 49.5014 |
| Image | **0.7522** | **6.0199** | 1.6129 | 10.2616 | 1.1499 | 14.3725 | 4.7013 | 19.3356 |
| Letter | **0.1712** | **7.4176** | 1.2489 | 7.4470 | 0.4427 | 8.7445 | 0.5743 | 15.1779 |
| Magic | **1.8314** | 3.9094 | 2.9405 | **3.3815** | 5.5790 | 4.2105 | 8.7810 | 9.0064 |
| Parkinsons | **0.3273** | 5.0597 | 0.8678 | **4.9044** | 3.3804 | 5.7260 | 18.3312 | 19.7001 |
| SkillCraft | **0.7669** | 8.2828 | 0.7771 | **8.2494** | 1.0283 | 9.9484 | 1.2849 | 15.9751 |
| Statlog | **0.0838** | **3.0998** | 0.3356 | 7.9734 | 0.4476 | 10.8263 | 13.8437 | 35.8268 |
| Steel | **1.1472** | 12.5944 | 1.2208 | **12.3096** | 4.8710 | 16.4015 | 3.8084 | 25.8953 |
| Taiwan Credit | **0.5523** | 10.9845 | 0.5710 | 10.9415 | 0.5744 | 13.0437 | 0.9535 | 21.8963 |
| Wine Quality | 0.6359 | **4.2801** | **0.3046** | 6.0936 | 1.5020 | 6.1118 | 3.0451 | 10.1001 |
| LFW | **0.4463** | **7.6229** | 0.5340 | 7.6361 | fail to converge | | | |

---

[6] The code to estimate this quantity is provided at the author's repository

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

## A PROOFS

### A.1 PROOFS OF SECTION 2

*Proof of Proposition 2.2.* Let $S = \mathbb{E}_{\mathbb{Q}}[XX^\top | A = 0] - \mathbb{E}_{\mathbb{Q}}[XX^\top | A = 1]$. We first prove the "only if" direction. Suppose that there exists a fair projection matrix $V \in \mathcal{M}_k$ relative to $\mathbb{Q}$. Let $U \in \mathcal{M}_{d-k}$ be a complement matrix of $V$. Then, Definition 2.1 can be rewritten as

$$\langle UU^\top, S \rangle = 0,$$

which implies that the null space of $S$ has a dimension at least $d - k$. By the rank-nullity duality, we have $\mathrm{rank}(S) \leq k$.

Next, we prove the "if" direction. Suppose that $\mathrm{rank}(S) \leq k$. Then, the matrix $S$ has at least $d - k$ (repeated) zero eigenvalues. Let $U \in \mathcal{M}_{d-k}$ be an orthonormal matrix whose columns are any $d - k$ eigenvectors corresponding to the zero eigenvalues of $S$ and $V \in \mathcal{M}_k$ be a complement matrix of $U$. Then,

$$\langle I_d - VV^\top, S \rangle = \langle UU^\top, S \rangle = 0.$$

Therefore, $V$ is a fair projection matrix relative to $\mathbb{Q}$. This completes the proof. $\qquad\square$

### A.2 PROOF OF SECTION 3

*Proofs of Proposition 3.3.* By exploiting the definition of the loss function $\ell$, we find

$$\sup_{\mathbb{Q}_a : W(\mathbb{Q}_a, \hat{\mathbb{P}}_a) \leq \varepsilon_a} \upsilon \mathbb{E}_{\mathbb{Q}_a}[\ell(V, X)]$$

$$= \begin{cases} \sup_{\mu_a, \Sigma_a} & \mathrm{tr}\left(\upsilon(I - VV^\top)(\Sigma_a + \mu_a \mu_a^\top)\right) \\ \mathrm{s.t.} & \|\mu_a - \hat{\mu}_a\|_2^2 + \mathrm{tr}\left(\Sigma_a + \hat{\Sigma}_a - 2\big(\hat{\Sigma}_a^{\frac{1}{2}} \Sigma_a \hat{\Sigma}_a^{\frac{1}{2}}\big)^{\frac{1}{2}}\right) \leq \varepsilon_a \end{cases}$$

$$= \begin{cases} \inf & \gamma(\varepsilon_a - \mathrm{tr}\left(\hat{\Sigma}_a\right)) + \gamma^2 \mathrm{tr}\left((\gamma I - \upsilon(I - VV^\top))^{-1}\hat{\Sigma}_a\right) + \tau \\ \mathrm{s.t.} & \begin{bmatrix} \gamma I - \upsilon(I - VV^\top) & \gamma \hat{\mu}_a \\ \gamma \hat{\mu}_a^\top & \gamma \|\hat{\mu}_a\|_2^2 + \tau \end{bmatrix} \succeq 0, \quad \gamma I \succ \upsilon(I - VV^\top), \quad \gamma \geq 0, \end{cases}$$

where the last equality follows from Nguyen (2019, Lemma 3.22). By the Woodbury matrix inversion, we have

$$(\gamma I - \upsilon(I - VV^\top))^{-1} = \gamma^{-1}I - \frac{\upsilon}{\gamma(\upsilon - \gamma)}(I - VV^\top).$$

Moreover, using the Schur complement, the semidefinite constraint is equivalent to

$$\gamma \|\hat{\mu}_a\|_2^2 + \tau \geq \gamma^2 \hat{\mu}_a^\top (\gamma I - \upsilon(I - VV^\top))^{-1}\hat{\mu}_a,$$

which implies that at optimality, we have

$$\tau = \frac{\upsilon\gamma}{\gamma - \upsilon}\hat{\mu}_a^\top (I - VV^\top)\hat{\mu}_a.$$

At the same time, the constraint $\gamma I \succ \upsilon(I - VV^\top)$ is equivalent to $\gamma > \upsilon$. Combining all previous equations, we have

$$\sup_{\mathbb{Q}_a : W(\mathbb{Q}_a, \hat{\mathbb{P}}_a) \leq \varepsilon_a} \upsilon \mathbb{E}_{\mathbb{Q}_a}[\ell(V, X)] = \inf_{\gamma > \max\{0, \upsilon\}} \gamma\varepsilon_a + \frac{\gamma\upsilon}{\gamma - \upsilon}\langle I_d - VV^\top, \hat{M}_a \rangle.$$

The dual optimal solution $\gamma^\star$ is given by

$$\gamma^\star = \begin{cases} \upsilon\left(1 + \sqrt{\dfrac{\langle I_d - VV^\top, \hat{M}_a \rangle}{\varepsilon_a}}\right) & \text{if } \upsilon \geq 0, \\ \upsilon\left(1 - \sqrt{\dfrac{\langle I_d - VV^\top, \hat{M}_a \rangle}{\varepsilon_a}}\right) & \text{if } \upsilon < 0 \text{ and } \langle I_d - VV^\top, \hat{M}_a \rangle \geq \varepsilon_a, \\ 0 & \text{if } \upsilon < 0 \text{ and } \langle I_d - VV^\top, \hat{M}_a \rangle < \varepsilon_a. \end{cases}$$

Note that $\gamma^\star \geq \max\{0, \upsilon\}$ in all the cases. Therefore, we have

$$\sup_{\mathbb{Q}_a : W(\mathbb{Q}_a, \hat{\mathbb{P}}_a) \leq \varepsilon_a} \upsilon \mathbb{E}_{\mathbb{Q}_a}[\ell(V, X)]$$

$$= \begin{cases} \upsilon\left(\sqrt{\varepsilon_a} + \sqrt{\langle I_d - VV^\top, \hat{M}_a \rangle}\right)^2 & \text{if } \upsilon \geq 0, \\ \upsilon\left(\sqrt{\varepsilon_a} - \sqrt{\langle I_d - VV^\top, \hat{M}_a \rangle}\right)^2 & \text{if } \upsilon < 0 \text{ and } \langle I_d - VV^\top, \hat{M}_a \rangle \geq \varepsilon_a, \\ 0 & \text{if } \upsilon < 0 \text{ and } \langle I_d - VV^\top, \hat{M}_a \rangle < \varepsilon_a. \end{cases}$$

This completes the proof. $\qquad\square$

We are now ready to prove Theorem 3.2.

*Proof of Theorem 3.2.* By expanding the absolute value, problem (4) is equivalent to

$$\min_{V \in \mathbb{R}^{d \times k}, V^\top V = I_k} \max\{J_0(V), J_1(V)\},$$

where for each $(a, a') \in \{(0,1),(1,0)\}$, we can re-express $J_a$ as

$$J_a(V) = \sup_{\mathbb{Q}_a : W(\mathbb{Q}_a, \hat{\mathbb{P}}_a) \leq \varepsilon_a} (\hat{p}_a + \lambda) \mathbb{E}_{\mathbb{Q}_a}[\ell(V, X)] + \sup_{\mathbb{Q}_{a'} : W(\mathbb{Q}_{a'}, \hat{\mathbb{P}}_{a'}) \leq \varepsilon_{a'}} (\hat{p}_{a'} - \lambda) \mathbb{E}_{\mathbb{Q}_{a'}}[\ell(V, X)]$$

Using Proposition 3.3 to reformulate the two individual supremum problems, we have

$$J_a(V) = (\hat{p}_a + \lambda)\varepsilon_a + 2|\hat{p}_a + \lambda|\sqrt{\varepsilon_a \langle I_d - VV^\top, \hat{M}_a \rangle} + (\hat{p}_a + \lambda)\langle I_d - VV^\top, \hat{M}_a \rangle$$

$$+ (\hat{p}_{a'} - \lambda)\varepsilon_{a'} + 2|\hat{p}_{a'} - \lambda|\sqrt{\varepsilon_{a'}\langle I_d - VV^\top, \hat{M}_{a'} \rangle} + (\hat{p}_{a'} - \lambda)\langle I_d - VV^\top, \hat{M}_{a'} \rangle.$$

By defining the necessary parameters $\kappa$, $\theta$, $\vartheta$ and $C$ as in the statement of the theorem, we arrive at the postulated result. $\qquad\square$

## A.3 PROOFS OF SECTION 4

*Proof of Lemma 4.1.* Let $a_U \in \arg\max_{a \in \{0,1\}} F_a(U)$ and $a'_U = 1 - a_U$. Then, an Euclidean subgradient of $F$ is given by

$$\nabla F(U) = \frac{\theta_{a_U}}{\sqrt{\langle UU^\top, \hat{M}_{a_U} \rangle}} \hat{M}_{a_U} U + \frac{\vartheta_{a'_U}}{\sqrt{\langle UU^\top, \hat{M}_{a'_U} \rangle}} \hat{M}_{a'_U} U + 2C_{a_U} U \in \mathbb{R}^{d \times (d-k)}.$$

The tangent space of the Stiefel manifold $\mathcal{M}$ at $U$ is given by

$$T_U \mathcal{M} = \{\Delta \in \mathbb{R}^{d \times (d-k)} : \Delta^\top U + U^\top \Delta = 0\},$$

whose orthogonal projection (Absil et al., 2007, Example 3.6.2) can be computed explicitly via

$$\text{Proj}_{T_U \mathcal{M}}(D) = (I_d - UU^\top)D + \frac{1}{2}U(U^\top D - D^\top U), \quad D \in \mathbb{R}^{d \times (d-k)}.$$

Therefore, a Riemannian subgradient of $F$ at any point $U \in \mathcal{M}$ is given by

$$\text{grad}F(U) = \text{Proj}_{T_U \mathcal{M}}(\nabla F(U))$$

$$= (I_d - UU^\top)\left(\frac{\theta_{a_U}}{\sqrt{\langle UU^\top, \hat{M}_{a_U}\rangle}}\hat{M}_{a_U}U + \frac{\vartheta_{a'_U}}{\sqrt{\langle UU^\top, \hat{M}_{a'_U}\rangle}}\hat{M}_{a'_U}U + 2C_{a_U}U\right).$$

In the last line, we have used the fact that, if $D = SU$ for some symmetric matrix $S$, then

$$U^\top D - D^\top U = U^\top S U - U^\top S^\top U = 0.$$

This completes the proof. $\qquad\square$

The proof of Lemma 4.2 relies on the following preliminary result.

**Lemma A.1.** Let $M \in \mathbb{R}^{(d-k) \times (d-k)}$ be a positive definite matrix. Then,

$$\left|\langle UU^\top, M \rangle - \langle U'U'^\top, M \rangle\right| \leq 2\sqrt{d-k}\sigma_{\max}(M)\|U - U'\|_F \quad \forall U, U' \in \mathcal{M}, \qquad (10)$$

and

$$\left|\sqrt{\langle UU^\top, M \rangle} - \sqrt{\langle U'U'^\top, M \rangle}\right| \leq \frac{\sigma_{\max}(M)}{\sqrt{\sigma_{\min}(M)}}\|U - U'\|_F \quad \forall U, U' \in \mathcal{M}, \qquad (11)$$

where $\sigma_{\max}(M)$ and $\sigma_{\min}(M)$ denote the maximum and minimum eigenvalues of the matrix $M$.

*Proof of Lemma A.1.* For inequality (10),

$$\left| \langle UU^\top, M \rangle - \langle U'U'^\top, M \rangle \right| \leq \left| \langle UU^\top, M \rangle - \langle UU'^\top, M \rangle \right| + \left| \langle UU'^\top, M \rangle - \langle U'U'^\top, M \rangle \right|$$
$$\leq \left| \langle U, M(U - U') \rangle \right| + \left| \langle U', M(U - U') \rangle \right|$$
$$\leq \|U\|_F \|M(U - U')\|_F + \|U'\|_F \|M(U - U')\|_F$$
$$= 2\sqrt{d - k}\,\sigma_{\max}\|U - U'\|_F.$$

For inequality (11), we first note that the function $x \mapsto \sqrt{x}$ is $1/(2\sqrt{x_{\min}})$-Lipschitz on $[x_{\min}, +\infty)$ and that

$$\langle UU^\top, M \rangle \geq (d - k)\sigma_{\min}(M) \quad \forall U \in \mathcal{M}.$$

Therefore,

$$\left| \sqrt{\langle UU^\top, M \rangle} - \sqrt{\langle U'U'^\top, M \rangle} \right| \leq \frac{1}{2\sqrt{(d - k)\sigma_{\min}(M)}} \left| \langle UU^\top, M \rangle - \langle U'U'^\top, M \rangle \right|$$
$$\leq \frac{\sigma_{\max}(M)}{\sqrt{\sigma_{\min}(M)}}\|U - U'\|_F,$$

where the last inequality follows from (10). This completes the proof. $\qquad\square$

We are now ready to prove Lemma 4.2.

*Proof of Lemma 4.2.* Let $U, U' \in \mathcal{M}$ be two arbitrary points. We have

$$|F(U) - F(U')|$$
$$= |\max\{F_0(U), F_1(U)\} - \max\{F_0(U'), F_1(U')\}|$$
$$\leq \max_{a \in \{0,1\}} |F_a(U) - F_a(U')|$$
$$\leq \max_{a \in \{0,1\}} \max\left\{ \theta_a \frac{\sigma_{\max}(\hat{M}_a)}{\sqrt{\sigma_{\min}(\hat{M}_a)}}, \vartheta_{1-a} \frac{\sigma_{\max}(\hat{M}_{1-a})}{\sqrt{\sigma_{\min}(\hat{M}_{1-a})}}, 2\sqrt{d - k}\,\sigma_{\max}(C_a) \right\} \|U - U'\|_F,$$

where the last inequality follows from the definition of $F_a$ and Lemma A.1. This completes the proof. $\qquad\square$

*Proof of Theorem 4.3.* The proof follows from the fact that $F$ is convex on the Euclidean space $\mathbb{R}^{d \times (d-k)}$, Lemma 4.2 and Li et al. (2021, Theorem 2) (and the remarks following it). $\qquad\square$

## B   EXTENSION TO NON-BINARY SENSITIVE ATTRIBUTES

The main paper focuses on the case of a binary sensitive attribute with $\mathcal{A} = \{0, 1\}$. In this appendix, we extend our approach to the case when the sensitive attribute is non-binary. Concretely, we suppose that the sensitive attribute $A$ can take on any of the $m$ possible values from 1 to $m$. In other words, the attribute space now becomes $\mathcal{A} = \{1, \ldots, m\}$.

**Definition B.1** (Generalized unfairness measure). The generalized unfairness measure is defined as the maximum pairwise unfairness measure, that is,

$$\mathbb{U}_{\max}(V, \mathbb{Q}) \triangleq \max_{(a,a') \in \mathcal{A} \times \mathcal{A}} |\mathbb{E}_{\mathbb{Q}}[\ell(V, X)|A = a] - \mathbb{E}_{\mathbb{Q}}[\ell(V, X)|A = a']|.$$

Notice that if $\mathcal{A} = \{0, 1\}$, then $\mathbb{U}_{\max} \equiv \mathbb{U}$ recovers the unfairness measure for binary sensitive attribute defined in Section 2.2. We now consider the following generalized fairness-aware PCA problem

$$\min_{V \in \mathbb{R}^{d \times k}, V^\top V = I_k} \sup_{\mathbb{Q} \in \mathbb{B}(\hat{\mathbb{P}})} \mathbb{E}_{\mathbb{Q}}[\ell(V, X)] + \lambda \mathbb{U}_{\max}(V, \mathbb{Q}). \tag{12}$$

Here we recall that the ambiguity set $\mathbb{B}(\hat{\mathbb{P}})$ is defined in (5). The next theorem provides the reformulation of (12).

**Theorem B.2** (Reformulation of non-binary fairness-aware PCA). Suppose that for any $a \in \mathcal{A}$, either of the following two conditions holds:

(i) Marginal probability bounds: $0 \leq \lambda \leq \hat{p}_a$,

(ii) Eigenvalue bounds: the empirical second moment matrix $\hat{M}_a = \frac{1}{N_a} \sum_{i \in \mathcal{I}_a} \hat{x}_i \hat{x}_i^\top$ satisfies $\sum_{j=1}^{d-k} \sigma_j(\hat{M}_a) \geq \varepsilon_a$, where $\sigma_j(\hat{M}_a)$ is the $j$-th smallest eigenvalues of $\hat{M}_a$.

Then problem (12) is equivalent to

$$
\min_{V \in \mathbb{R}^{d \times k}, V^\top V = I_k} \max_{a \neq a'} \left\{ \sum_{b \in \mathcal{A}} 2c_{a,a',b} \sqrt{\varepsilon_b \langle I_d - VV^\top, \hat{M}_b \rangle} + \lambda \langle I_d - VV^\top, \hat{M}_a - \hat{M}_{a'} \rangle + \lambda(\varepsilon_a - \varepsilon_{a'}) \right\},
$$

where the parameter $c_{a,a',b}$ admits values

$$
c_{a,a',b} = \begin{cases} \hat{p}_a + \lambda & \text{if } b = a, \\ |\hat{p}_{a'} - \lambda| & \text{if } b = a', \\ \hat{p}_b & \text{otherwise.} \end{cases}
$$

*Proof of Theorem B.2.* For simplicity, we let $E(V, \mathbb{Q}, b) = \mathbb{E}_\mathbb{Q}[\ell(V, X) | A = b]$. Then, the objective function of problem (12) can be re-written as

$$
\sup_{\mathbb{Q} \in \mathbb{B}(\hat{\mathbb{P}})} \mathbb{E}_\mathbb{Q}[\ell(V, X)] + \lambda \mathbb{U}_{\max}(V, \mathbb{Q})
$$

$$
= \sup_{\mathbb{Q} \in \mathbb{B}(\hat{\mathbb{P}})} \sum_{b \in \mathcal{A}} \hat{p}_b E(V, \mathbb{Q}, b) + \lambda \max_{a \neq a'} \{E(V, \mathbb{Q}, a) - E(V, \mathbb{Q}, a')\}
$$

$$
= \max_{a \neq a'} \left\{ \sum_{b \neq a, a'} \sup_{W(\mathbb{Q}_b, \hat{\mathbb{P}}_b) \leq \varepsilon_b} \hat{p}_b E(V, \mathbb{Q}_b, b) + \sup_{W(\mathbb{Q}_a, \hat{\mathbb{P}}_a) \leq \varepsilon_a} (\hat{p}_a + \lambda) E(V, \mathbb{Q}_a, a) + \sup_{W(\mathbb{Q}_{a'}, \hat{\mathbb{P}}_{a'}) \leq \varepsilon_{a'}} (\hat{p}_{a'} - \lambda) E(V, \mathbb{Q}_{a'}, a') \right\}
$$

$$
= \max_{a \neq a'} \left\{ \sum_{b \neq a, a'} \hat{p}_b \left( \sqrt{\langle I_d - VV^\top, \hat{M}_b \rangle} + \sqrt{\varepsilon_b} \right)^2 + (\hat{p}_a + \lambda) \left( \sqrt{\langle I_d - VV^\top, \hat{M}_a \rangle} + \sqrt{\varepsilon_a} \right)^2 \right.
$$

$$
\left. + (\hat{p}_{a'} - \lambda) \left( \sqrt{\langle I_d - VV^\top, \hat{M}_{a'} \rangle} + \mathrm{sgn}(\hat{p}_{a'} - \lambda)\sqrt{\varepsilon_{a'}} \right)^2 \right\}
$$

$$
= \max_{a \neq a'} \left\{ \sum_{b \in \mathcal{A}} \hat{p}_b \left( \langle I_d - VV^\top, \hat{M}_b \rangle + \varepsilon_b \right) + \sum_{b \in \mathcal{A}} 2c_{a,a',b} \sqrt{\varepsilon_b \langle I_d - VV^\top, \hat{M}_b \rangle} \right.
$$

$$
\left. + \lambda \left( \langle I_d - VV^\top, \hat{M}_a - \hat{M}_{a'} \rangle + \varepsilon_a - \varepsilon_{a'} \right) \right\},
$$

where the first equality follows from the definition of $\mathbb{U}_{\max}(V, \mathbb{Q})$ and $E(V, \mathbb{Q}, b)$, the second from the definition (5) of the ambiguity set $\mathbb{B}(\hat{\mathbb{P}})$, the third from Proposition 3.3 and the fourth from the definition of $c_{a,a',b}$. Noting that the first sum in the above maximization is independent of $a$ and $a'$, the proof is completed. ∎

Theorem 12 indicates that if the sensitive attribute admits finite values, then the distributionally robust fairness-aware PCA problem using an $\mathbb{U}_{\max}$ unfairness measure can be reformulated as an optimization problem over the Stiefel manifold, where the objective function is a pointwise maximization of finite number of individual functions. It is also easy to see that each individual function can be reparametrized using $U$, and the Riemannian gradient descent algorithm in Section 4 can be adapted to solve for the optimal solution. The details on the algorithm are omitted.

## C  INFORMATION ON DATASETS

We summarize here the number of observations, dimensions, and the sensitive attribute of the data sets. For further information about the data sets and pre-processing steps, please refer to Samadi et al. (2018) for Default Credit and Labeled Faces in the Wild (LFW) data sets, and Olfat & Aswani (2019) for others. For each data set, we further remove columns with too small standard deviation ($\leq 1e^{-5}$) as they do not significantly affect the results, and ones with too large standard deviation ($\geq 1000$) which we consider as unreliable features.

Table 2: Number of observations $N$, dimensions $d$, and sensitive attribute $A$ of datasets used in this paper. (y - yes, n - no)

|  | Default Credit | Biodeg | E. Coli | Energy | German Credit |
|---|---|---|---|---|---|
| $N$ | 30000 | 1055 | 333 | 768 | 1000 |
| $d$ | 22 | 40 | 7 | 8 | 48 |
| $A$ | Education (high/low) | Ready Biodegradable (y/n) | isCytoplasm (y/n) | Orientation$< 4$ (y/n) | $A13 \geq 200$DM (y/n) |
|  | Image | Letter | Magic | Parkinsons | SkillCraft |
| $N$ | 660 | 20000 | 19020 | 5875 | 3337 |
| $d$ | 18 | 16 | 10 | 20 | 17 |
| $A$ | class (path/grass) | Vowel (y/n) | classIsGamma (y/n) | Sex (male/female) | Age$> 20$ (y/n) |
|  | Statlog | Steel | Taiwan Credit | Wine Quality | LFW |
| $N$ | 3071 | 1941 | 29623 | 6497 | 4000 |
| $d$ | 36 | 24 | 22 | 11 | 576 |
| $A$ | RedSoil (vsgrey/dampgrey) | FaultOther (y/n) | Sex (male/female) | isWhite (y/n) | Sex (male/female) |

## D  ADDITIONAL RESULTS

### D.1  DETAIL PERFORMANCES

Table 3 shows the performances of four examined methods with two criteria ABDiff. and ARE. It is clear that our method achieves the best results over all 14 datasets w.r.t. ABDiff., and 7 datasets on ARE., which is equal to the number of datasets FairPCA out-perform others.

Table 4 complements Table 1 from the main text, from which we can see that two versions of CFPCA out-perform others over all datasets w.r.t. $\triangle \mathcal{F}_{Lin}$, which is the criteria they optimize for.

Table 3: In-sample performance over two criteria

| | RFPCA | | FairPCA | | CFPCA-Mean Con. | | CFPCA - Both Con. | |
|---|---|---|---|---|---|---|---|---|
| Dataset | ABDiff. | ARE. | ABDiff. | ARE. | ABDiff. | ARE. | ABDiff. | ARE. |
| Default Credit | **0.9457** | 9.9072 | 1.5821 | **9.9049** | 0.9949 | 10.5164 | 3.2827 | 21.4523 |
| Biodeg | **9.4093** | **23.1555** | 14.2587 | 23.8227 | 15.5545 | 26.6540 | 24.8706 | 39.8737 |
| E. Coli | **0.5678** | **1.4804** | 0.9191 | 2.0840 | 0.9539 | 2.8360 | 4.5225 | 5.2155 |
| Energy | **0.0094** | 0.2295 | 0.0153 | **0.2273** | 0.2658 | 2.7893 | 0.2136 | 7.8768 |
| German Credit | **1.6265** | **40.1512** | 2.9824 | 40.3393 | 2.6109 | 40.1860 | 2.8741 | 47.1006 |
| Image | **0.1320** | **5.0924** | 0.7941 | 9.0437 | 0.6910 | 13.4491 | 3.0118 | 18.0000 |
| Letter | **0.1121** | **7.4088** | 1.2560 | 7.4375 | 0.4572 | 8.7764 | 0.5301 | 15.2234 |
| Magic | **1.7405** | 3.8766 | 2.8679 | **3.3500** | 5.5405 | 4.1938 | 8.7963 | 8.9695 |
| Parkinsons | **0.1238** | 5.0471 | 0.6702 | **4.8760** | 3.9470 | 5.9379 | 17.8122 | 19.9788 |
| SkillCraft | **0.4231** | 8.1569 | 0.5576 | **8.1096** | 0.7156 | 9.7755 | 0.9334 | 15.8245 |
| Statlog | **0.1972** | **3.0588** | 0.3315 | 7.9980 | 0.3857 | 10.9358 | 13.0725 | 35.9214 |
| Steel | **0.6943** | 11.0396 | 1.8015 | **10.7653** | 2.8933 | 14.5680 | 1.9322 | 23.9906 |
| Taiwan Credit | **1.1516** | 10.5136 | 1.3362 | **10.4478** | 1.3158 | 12.5867 | 2.2720 | 21.4365 |
| Wine Quality | **0.1125** | **4.1491** | 0.1705 | 5.8999 | 1.1359 | 5.9117 | 2.5852 | 9.8959 |
| LFW | **0.4147** | 7.5137 | 0.5300 | **7.5127** | fail to converge | | | |

**Adjustment for the LFW dataset.** To demonstrate the efficacy of our method on high-dimensional data sets, we also do experiments on a subset of 2000 faces for each of male and female group (4000

Table 4: Out-of-sample performance measured using the $\triangle \mathcal{F}_{Lin}$ criterion.

| | RFPCA | FairPCA | CFPCA-Mean Con. | CFPCA - Both Con. |
|---|---|---|---|---|
| Default Credit | 0.1596 | 0.2236 | 0.0574 | **0.0413** |
| Biodeg | 0.4892 | 0.4759 | 0.2014 | **0.1371** |
| E. Coli | 0.8556 | 0.7444 | 0.4455 | **0.2532** |
| Energy | 0.0580 | 0.0554 | **0.0502** | 0.0736 |
| German Credit | 0.1997 | 0.1737 | 0.1408 | **0.1093** |
| Image | 0.9996 | 0.9498 | **0.1874** | 0.2013 |
| Letter | 0.0954 | 0.0942 | 0.0556 | **0.0455** |
| Magic | 0.2195 | 0.2531 | 0.1561 | **0.0882** |
| Parkinson's | 0.1459 | 0.1061 | 0.1805 | **0.0480** |
| SkillCraft | 0.1126 | 0.1141 | **0.0721** | 0.0742 |
| Statlog | 0.9804 | 0.6309 | 0.1359 | **0.0669** |
| Steel | 0.2288 | 0.2240 | 0.1418 | **0.0875** |
| Taiwan Credit | 0.0604 | 0.0535 | 0.0391 | **0.0370** |
| Wine Quality | 0.9699 | 0.4639 | 0.2192 | **0.0817** |

in total) from LFW dataset,[7] all images are rescaled to resolution $24 \times 24$ (dimensions $d = 576$). The experiment follows the same procedure in Section 5, with reducing the number of iterations to $500$ for both `RFPCA` and `FairPCA` and 2-fold cross validation, the results are averaged over 10 train-test simulations. Due to the high dimension of the input, the implementation of Olfat & Aswani (2019) fails to return any result.

---

[7] https://github.com/samirasamadi/Fair-PCA

## D.2 VISUALIZATION

### D.2.1 EFFECTS OF THE AMBIGUITY SET RADIUS

We examine the change of the model's performance with respect to the change of the radius of the ambiguity sets. To generate the toy data (also used for Figure 1), we use two 2-dimensional Gaussian distributions to represent two groups of a sensitive attribute, $A = 0$ and $A = 1$, or groups 0 and 1 for simplicity. The two distributions both have the mean at $(0, 0)$ and covariance matrices for group 0 and 1 are

$$\begin{pmatrix} 4.0 & 0 \\ 0 & 0.2 \end{pmatrix} \quad \text{and} \quad \begin{pmatrix} 0.2 & 0.4 \\ 0.4 & 3.0 \end{pmatrix},$$

respectively. For the test set, the number of samples is 8000 for group 0 and 4000 for group 1, while for the training set, we have 200 for group 0 and 100 for group 1. We average the results over 100 simulations, for each simulation, the test data is fixed, the training data is randomly generated with the number of samples mentioned above. The projections are learned on training data and measured on test data by the summation of ARE. and ABDiff. We fixed $\lambda = 0.1$, which is not too small for achieving fair projections, and not too large to clearly observe the effects of $\varepsilon$, and we also fixed $\varepsilon_0$ for better visualization. Note that we still compute $\varepsilon_1 = \alpha/\sqrt{N_1}$ in which, $\alpha$ is tested with 100 values evenly spaced in $[0, 10]$.

The experiment results are visualized in Figure 4. The result suggests that increasing the ambiguity set radius can improve the overall model's performance. This justifies the benefit of adding distributional robustness to the fairness-aware PCA model. After a saturation point, a too large radius can lessen the role of empirical data, and the model prioritizes a more extreme distribution that is far from the target distribution, which causes the reduction in the model's performance on target data.

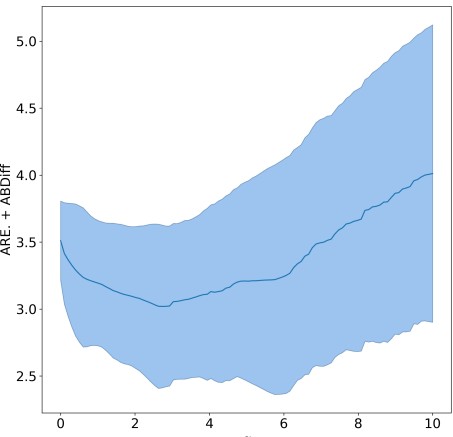

Figure 4: Performance changes w.r.t. the ambiguity set's radius. The solid line is the average over 100 simulations, and the shade represent the 1-standard deviation range.

### D.2.2 PARETO CURVES

Figures 5 and 6 plot the Pareto frontier for two datasets (Biodeg and German Credit) with 3 principal components. One can observe that `RFPCA` produces points that dominate other methods based on the trade-off between ARE. and ABDiff.

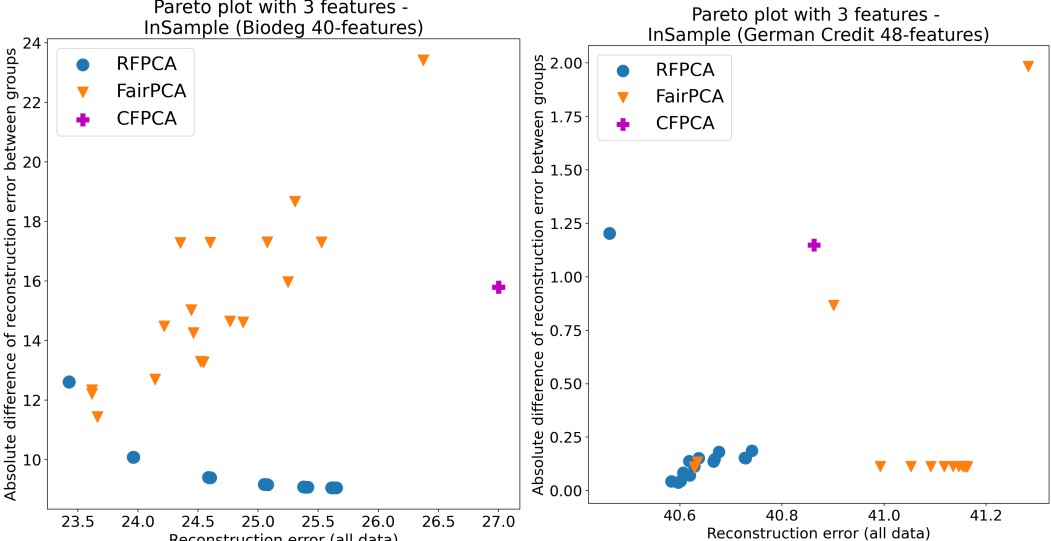

Figure 5: Pareto curves on Biodeg dataset (all data) with 3 principal components

Figure 6: Pareto curves on German Credit dataset (all data) with 3 principal components

### D.2.3 PERFORMANCE WITH DIFFERENT PRINCIPAL COMPONENTS

We collect here the reconstruction errors for different numbers of principal components.

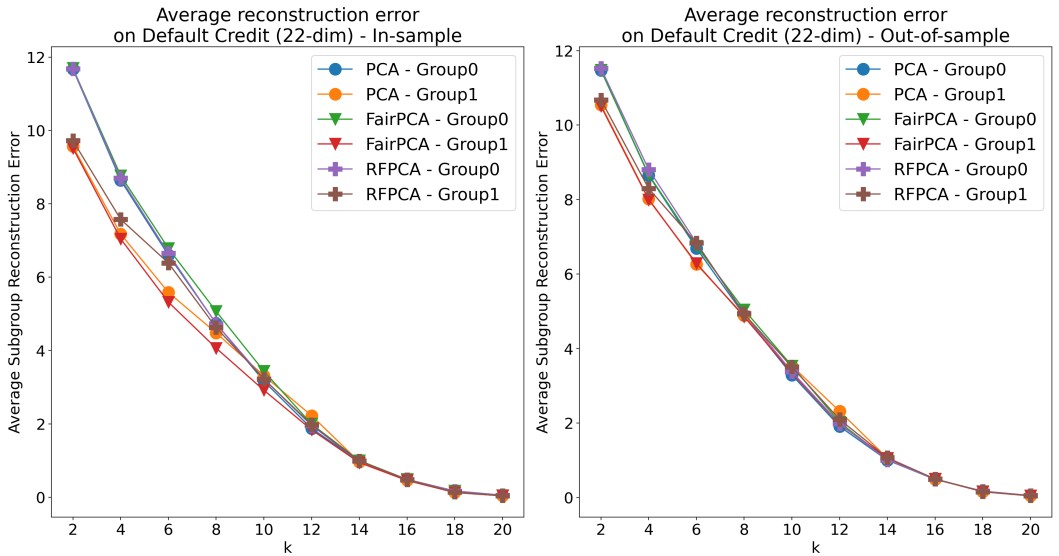

Figure 7: Subgroup average error
with different $k$ on Default Credit dataset

Figure 8: Subgroup average error
with different $k$ on Default Credit dataset

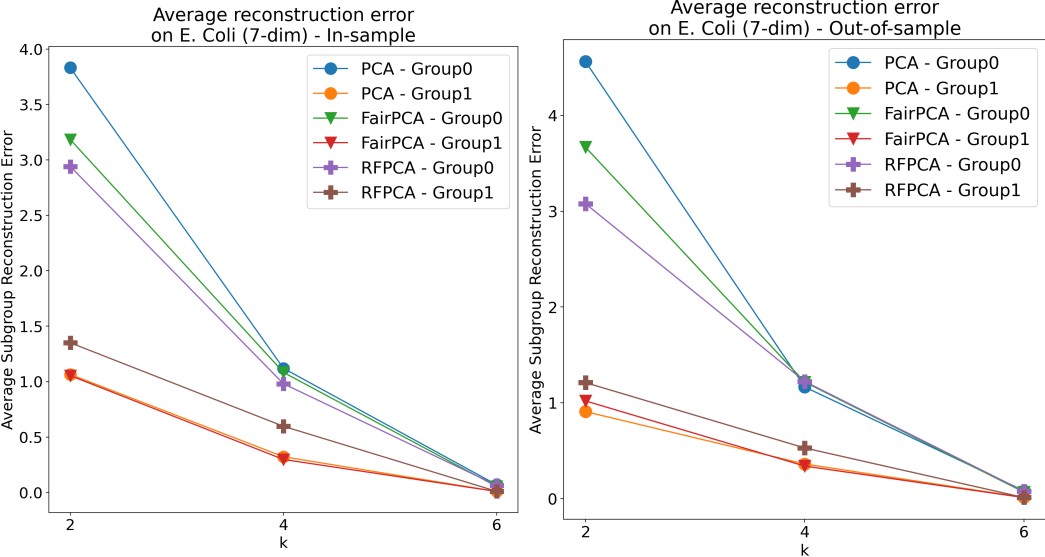

Figure 9: Subgroup average error
with different $k$ on E. Coli dataset

Figure 10: Subgroup average error
with different $k$ on E. Coli dataset

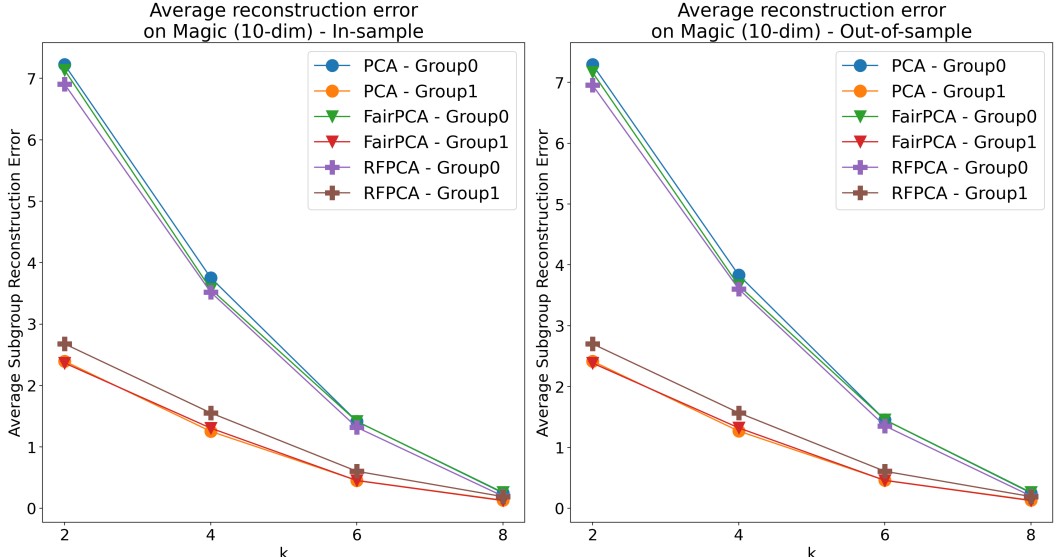

Figure 11: Subgroup average error with different $k$ on Magic dataset

Figure 12: Subgroup average error with different $k$ on Magic dataset

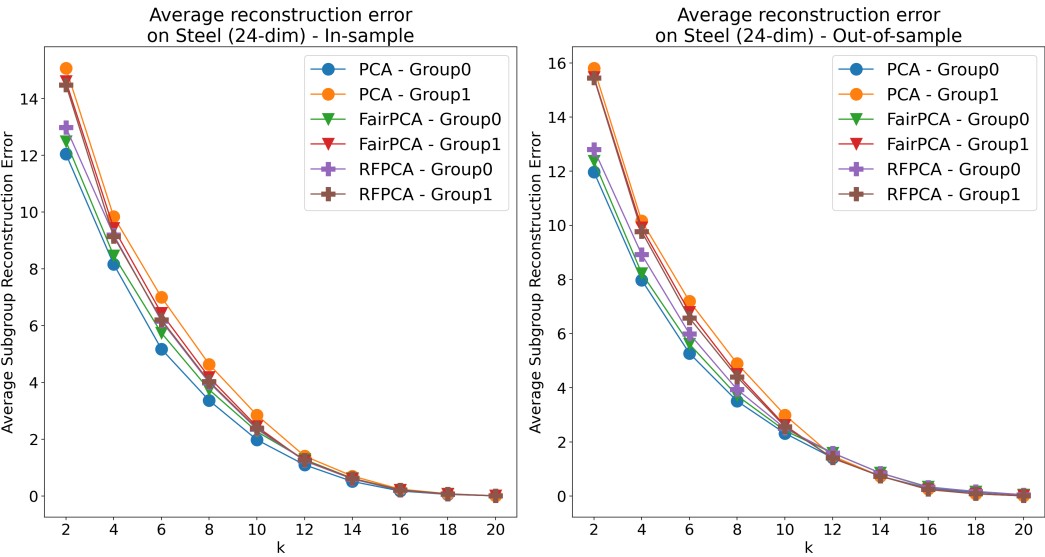

Figure 13: Subgroup average error with different $k$ on Steel dataset

Figure 14: Subgroup average error with different $k$ on Steel dataset

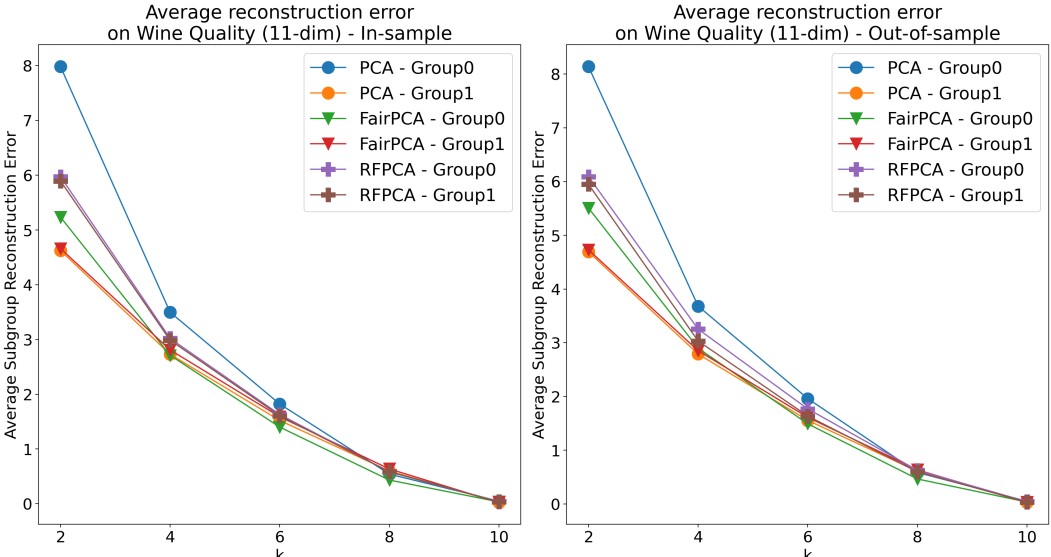

Figure 15: Subgroup average error
with different $k$ on Wine Quality dataset

Figure 16: Subgroup average error
with different $k$ on Wine Quality dataset

