# OpenReview forum: "Distributionally Robust Fair Principal Components via Geodesic Descents"
_ICLR.cc/2022/Conference — ICLR 2022 Poster_

### Official Review · Reviewer_8Bm5 · 2021-11-02

**Correctness:** 3
**Technical Novelty And Significance:** 3
**Empirical Novelty And Significance:** 2
**Recommendation:** 6
**Confidence:** 4

**Main Review:**

The strength of the paper is the rigorous optimization framework to solve the proposed fair PCA, with a nice reformulation and theoretical property.

The weakness of this paper is in the following folds. First, it is not very clear on the significance of the novelty. The authors cast the PCA problem as the minimization of the penalized reconstructed error. Such a formulation is widely common in the machine learning community. Because of the L2-type loss function, the use of the distributionally robust optimization is not that surprising with the known complication on the penalty part. Second, the proposed method involves the selection of tuning parameter. While the authors seem to avoid the discussion tuning parameter selection. Moreover, the numerical experiments just consider the use of fixed value of these tuning parameters, which makes reader hard to judge the quality of the numerical comparison. Third, due the use of the penalty to engage the fairness, it is not how good the proposed method enables the fairness, especially there is no criterion on the selection of the tuning parameter \lambda. Fourth, under the context of fairness, the authors also avoid the discussion on how the choice of k, the number of PCs, would affect the fairness.

**Summary Of The Paper:**

This paper considers the fair PCA with distribution ally robust optimization algorithm. The problem is of great interest. The key idea of the proposed method is to consider a penalized loss function with the penalty on the fairness criterion. The distributionally robust optimization and its reformulation is developed for parameter estimation.

**Summary Of The Review:**


Based on the aforementioned strengths and weakness, I would consider the proposed method contain interest merits, with the conservation of the sufficiency on the methodology novelty, framework rigor, and the applicability in practice.

---

> ### Author Response · Authors · 2021-11-22
> **Response to Reviewer 8Bm5 (1/2)**
>
> We thank the reviewer for raising interesting concerns about our work, please find below the detailed reply to your comments.
>
> **Q: First, it is not very clear on the significance of the novelty. The authors cast the PCA problem as the minimization of the penalized reconstructed error. Such a formulation is widely common in the machine learning community. Because of the L2-type loss function, the use of the distributionally robust optimization is not that surprising with the known complication on the penalty part**
>
> Our paper blends ideas and techniques from two prominent fields of machine learning: (distributional) robustness and fairness. Nevertheless, we do not agree with the statement that our results are “natural extensions”. On the contrary, our results require many non-trivial designs which we elaborate below:
> - A separable ambiguity set: The ambiguity set $\mathbb B (\hat{\mathbb P})$ is constructed from individual ambiguity sets on the space of *conditional* probability distributions. Only by using this construct can we decompose the objective function into a pointwise maximum of two individual functions $J_0(V)$ and $J_1(V)$ which admit analytical reformulations.
> Note that in the existing literature of distributionally robust optimization, the ambiguity set is constructed on the *joint* distribution. Using a joint ambiguity set will not allow such decomposition.
> - Wasserstein-type distance: The Wasserstein-type distance is compulsory in order to obtain the reformulation in Proposition 3.3. If *any* other distance or divergence is used, then one may not be able to arrive at closed-form reformulations. We would like to emphasize that the closed-form expression of Proposition 3.3 is crucial for algorithmic analysis in Section 4. Thus, the choice of the Wasserstein-type distance has been decided with a specific purpose of facilitating the Riemannian geodesic descent algorithm. Moreover, Proposition 3.3, which is the main tool we used in our reformulation results, is new and does not follow from any of the existing results in the distributionally robust optimization literature.
> - Reparametrization: The reformulation with the variable $V$ is not convex, and we require an additional step to reparametrize the resulting problem in the complementary variable $U$ to obtain a convex problem. This reparametrization, despite being simple, is also crucial. Moreover, existing literature on optimization on the Stiefel manifold often makes a direct assumption on convexity (i.e., convex in $V$). Thus, on the modelling viewpoint, this complementary reparametrization is critical to resolve the convexity issue.
> - Condition verification: Our paper also verifies the conditions of convergence by providing an explicit formula for the Lipschitz constant in Lemma 4.2.
>
> With the above four points, we would like to emphasize that the results of this paper are non-trivial. In contrast, these results are the culmination of many appropriate modelling choices. If any of the above four points are removed or altered, the final results will no longer hold.

---

> > ### Author Response · Authors · 2021-11-23
> > **Response to Reviewer 8Bm5 (2/2)**
> >
> > **Q: Second, the proposed method involves the selection of tuning parameter. While the authors seem to avoid the discussion tuning parameter selection. Moreover, the numerical experiments just consider the use of fixed value of these tuning parameters, which makes reader hard to judge the quality of the numerical comparison. Third, due the use of the penalty to engage the fairness, it is not how good the proposed method enables the fairness, especially there is no criterion on the selection of the tuning parameter $\lambda$.**
> >
> > We want to clarify that our experiment performs a proper cross-validation process to obtain the results in Table 1. The cross-validation procedure is necessary in the experiment leading to Table 1 because we are comparing the performance of our method against SOTA methods (Samadi et al. (2018) and Olfat & Aswani (2019)).
> >
> > In the experiment leading to Figure 3, we fix the hyper-parameters when checking the performance of the methods over different number of principal components $k$. Fixing the hyper-parameters is necessary because we are interested in observing the effect of the number of PCs $k$ to the fairness of models.
> >
> > Regarding choosing $\lambda$, it is clear from Figure 2 that there is a trade-off between the total reconstruction error and the fairness level. Then, the optimal $\lambda$ really depends on practitioners’ preference. Furthermore, Figure 1 can capture the intuition of the efficacy of our method in enabling the fairness of the PCA model: by tuning the value $\lambda$, the practitioners can obtain a spectrum of solutions that can better meet their preferences than existing methods such as Samadi et al. (2018).
> >
> > **Q: Fourth, under the context of fairness, the authors also avoid the discussion on how the choice of k, the number of PCs, would affect the fairness**
> >
> > It is true that the number of PCs plays an important role in the degree of (un)fairness. Our paper does not aim to prove any theoretical results connecting $k$ and (un)fairness. On the contrary, our experiments shed lights on the empirical connection between these two quantities.
> >
> > In Figure 3, we can observe the empirical fairness (measured by the gap between the group reconstruction errors) and the number of PCs. We can observe that increasing the number of PCs can reduce the effect of unfairness as the gap between subgroups’ reconstruction error reduces (for both fairness-aware models and the traditional model). Moreover, the error decrease does not follow a linear relationship in $k$. This trend is reinforced by more visualizations in Appendix D.2.3 of how the model’s performance changes with different numbers of PCs on various datasets.

---

### Official Review · Reviewer_wgFH · 2021-11-02

**Correctness:** 3
**Technical Novelty And Significance:** 3
**Empirical Novelty And Significance:** 3
**Recommendation:** 6
**Confidence:** 3

**Main Review:**

This paper tackles an interesting problem, and is mostly well-written and easy to follow. The problem formulation and re-formulation steps are not trivial.

However, there are places in this paper that needs more explanation and justification.

- There needs to be a discussion on the assumptions for Theorem 3.2. How restrictive are they? Were they satisfied in the experiments? For example, (i) implies that the regularization parameter is less than 0.5, which will restrict the flexibility shown in Figure 1.
- The ambiguity set B(P) is a central part of the proposed method. What is the justification for using Wasserstein-type divergence as opposed to others? Also, the second line of equation (5) needs to be explained more carefully.
- In defining $gap_\mu(U)$, is $||U'-U||_F$ the best choice to measure the difference between two orthonormal matrices? One can think of many examples with rotations where $U$ and $U'$ span the same subspace but will have very large frobenius norm difference. A more typical choice in the literature would be $||UU^T - U' U' ^T||$.
- Is the RHS of Theorem 4.3. necessarily decreasing with $t$? It's not obvious to me why this guarantees convergence.
- Experiments and discussions on the robustness of the proposed method are lacking, compared to the fairness aspect. Are there any comparisons to other robust PCA methods, or simulation experiments?
- In the experiments, how were the sensitive attributes chosen? Were they based on what are considered sensitive attributes in the real world, e.g. race, gender, or were they chosen randomly?
- This paper is lacking a conclusion section. It would have made it stronger if the authors mentioned how they would extend this work to non-binary sensitive attributes.
- On a very high level, it's not *as* obvious why reconstruction error should be balanced compared to, for example, in supervised learning. The toy example in Figure 1 is helpful but I'm wondering if the authors can describe a real-world example where bias in reconstruction error will be harmful - where PCA is the end result rather than a preprocessing step for supervised learning problem.

Other minor problems that can be fixed with careful editing, such as:
- Page 2: "signal process" -> signal processing
- Page 5: "By the definition the ambiguity..."
- Page 7: "At each iteration, the point..." sentence is hard to parse.
- Page 9: "In Each dataset, ..." "other features is..." "the the generalization..."

**Summary Of The Paper:**

This paper aims to improve fairness and distributional robustness in dimensionality reduction techniques. More specifically, it proposes a regularization term to principal compoenent analysis (PCA) such that the expected reconstruction errors for each groups (conditioned on a binary sensitive variable) are similar, and relaxes constraint on the empirical probability distribution by maximizing over a Wasserstein-type ambiguity set. This is optimized with subgradient descent on the Stiefel manifold, and numerical experiments comparing the proposed method to previous fair PCA are conducted.

**Summary Of The Review:**

It's a good paper, but is missing some interpretation of its theoretical results and justifications. It is likely that I will change my recommendation after the discussion period, if the authors provide more details.

---

> ### Author Response · Authors · 2021-11-22
> **Response to Reviewer wgFH (1/3)**
>
> We thank the reviewer for a careful read of our manuscript. Please find the detailed answer to your questions below.
>
> **Q: There needs to be a discussion on the assumptions for Theorem 3.2. How restrictive are they? Were they satisfied in the experiments? For example, (i) implies that the regularization parameter is less than 0.5, which will restrict the flexibility shown in Figure 1.**
>
> The assumption of Theorem 3.2 is actually very mild as it can always be satisfied by choosing a small enough regularization parameter $\lambda$ or a small enough radius $\varepsilon$. Note that the conditions are of the form “either… or…”, thus by satisfying one of them, we can freely vary the other.
>
> **Q: The ambiguity set B(P) is a central part of the proposed method. What is the justification for using Wasserstein-type divergence as opposed to others? Also, the second line of equation (5) needs to be explained more carefully.**
>
>
> We concur with the reviewer that the ambiguity set $\mathbb B(\hat{\mathbb P})$ plays an important role in the development of our paper. We highlight the below two important properties of this ambiguity set that are useful in this paper:
> - A separable ambiguity set: The ambiguity set $\mathbb B(\hat{\mathbb P})$ is constructed from individual ambiguity sets on the space of *conditional* probability distributions. Only by using this construct can we decompose the objective function into a pointwise maximum of two individual functions $J_0(V)$ and $J_1(V)$.
> Note that in the existing literature of distributionally robust optimization, the ambiguity set is constructed on the *joint* distribution. Using a joint ambiguity set will now allow such decomposition.
> - Wasserstein-type distance: The Wasserstein-type distance is compulsory in order to obtain the reformulation in Proposition 3.3. If *any* other distance or divergence is used, then one may not be able to arrive at closed-form reformulations. We would like to emphasize that the closed-form expression of Proposition 3.3 is crucial for algorithmic analysis in Section 4. Thus, the choice of the Wasserstein-type distance has been hand-crafted with a specific purpose of facilitating the Riemannian geodesic descent algorithm.
>
> The second line in equation (5) requires that the distribution $\mathbb{Q}$ can be written using the law of conditional probability with the conditional distributions $\mathbb Q_a$. In our first submission, we have used the measure-theoretic formula for the law of conditional probability, which is
> \\[
> \mathbb Q (\mathrm{d} x \times \mathrm{d} a) = \sum_{a \in \mathcal A} \hat p_{a} \mathbb Q_{a}(\mathrm{d} x) \delta_{a} (\mathrm{d}a).
> \\]
>
> In the revision, we have revised this equation into a form that is easier to parse in terms of the marginal distribution $\hat p_a$ and the conditional measure $\mathbb{Q}_a$:
> \\[
> \mathbb{Q}(\mathbb X \times \{a\} ) = \hat p_a \mathbb{Q}_a(\mathbb X) \quad \forall \mathbb X \subseteq \mathbb R^d,~a \in \mathcal A.
> \\]
>
>
> **Q: In defining  $gapμ(U)$, is $||U’−U||_F$  the best choice to measure the difference between two orthonormal matrices? One can think of many examples with rotations where $U$ and $U’$  span the same subspace but will have very large frobenius norm difference. A more typical choice in the literature would be $||UUT−U’U’^\top ||$**
>
> We agree with the reviewer that it also suffices to bound $\| UU^\top - U’ {U’}^\top \|_F$. However, the theory enjoyed by our algorithm is based on the stronger measure of $\|U - U’\|_F$, and hence would imply a bound for the quantity $\| UU^\top - U’ {U’}^\top \|_F$. Indeed, the Stiefel manifold is a compact set and the quadratic function $UU^\top$ is Lipschitz on any compact set. Therefore, there exists a constant $c>0$ such that $\| UU^\top - U’ {U’}^\top \|_F \le c  \| U - U’ \|_F$.
>
> **Q: Is the RHS of Theorem 4.3. necessarily decreasing with t? It's not obvious to me why this guarantees convergence.**
>
> Yes, the RHS is decreasing. The quantity $\tau$ is the number of iterations. As $\tau$ becomes larger, the RHS converges to 0.

---

> > ### Author Response · Authors · 2021-11-22
> > **Response to Reviewer wgFH (2/3)**
> >
> > **Q: Experiments and discussions on the robustness of the proposed method are lacking, compared to the fairness aspect. Are there any comparisons to other robust PCA methods, or simulation experiments?**
> >
> > Most of the existing robust PCA methods focus on mitigating the effects of corrupted data/noise on the reconstruction error and neglect the fairness criteria. There is so far no clear reason to believe that existing robust PCA methods may improve fairness. It is thus not clear to us why a comparison versus other robust PCA methods are necessary.
> >
> > We, however, agree that a better demonstration on the benefit of distributional robustness is desirable, which can be demonstrated more easily using a toy example with two dimensions. We use two $2$-dimensional Gaussian distributions to represent two groups of a sensitive attribute, two distributions both have the mean at $(0,0)$ and covariance matrices are \\[\begin{pmatrix}  4.0 & 0 \\\  0 & 0.2 \end{pmatrix} \quad \text{and} \quad \begin{pmatrix}  0.2 & 0.4 \\\  0.4 & 3.0 \end{pmatrix},\\]  respectively. For each simulation, we independently generate $200$ samples for the majority group and $100$ samples for the minority group. The projections are learned on training data and measured on a fixed test dataset consisting of $8000$ samples from the majority, and $4000$ samples from the minority group. We measure the weighted summation of $\text{ARE.}$~and $\text{ABDiff.}$ with $\lambda=0.1$, and the result is averaged over $100$ simulations. Suppose that group $1$ is the minority group, we use $\varepsilon_1 = \alpha$ in which, $\alpha$ is tested with $100$ values evenly spaced in $[0, 1]$.
> > The average loss collected from the experiment indicates that a value $\alpha > 0$ also minimizes the weighted error from the test dataset. This observation once again strengthens the needs for distributional robustness in the formulation. The detailed experiments are included in Appendix D.
> >
> >
> >
> > **Q: In the experiments, how were the sensitive attributes chosen? Were they based on what are considered sensitive attributes in the real world, e.g. race, gender, or were they chosen randomly?**
> >
> > The sensitive attributes are taken from existing papers on fair PCA. More specifically, “Default Credit” and “LFW” datasets are provided by Samadi et al. (2018) and the others are provided by Olfat & Aswani (2019). We added information about sensitive attributes in Appendix C in the revision, which is summarized in the table below ($A$ is the sensitive attribute, y-yes, n-no)
> >
> > | Dataset       | Default Credit   | Biodeg                | E. Coli       | Energy            | German Credit |
> > | :---- |  :----          | :----                | :----        | :----            | :----        |
> > |$A$     | Education (high/low)        | Ready Biodegradable (y/n)  | isCytoplasm (y/n)  | Orientation$<4$ (y/n)  | $A13 \ge 200 \text{DM}$ (y/n)        |
> >
> > |   Dataset      | Image            | Letter | Magic        | Parkinsons | SkillCraft   |
> > | :---- |  :----            | :---- | :----       | :----     | :----       |
> > |$A$     | class (path/grass) | Vowel (y/n) | classIsGamma (y/n)| Sex (male/female)        | Age$>20$ (y/n)    |
> >
> > |    Dataset     | Statlog                  | Steel         | Taiwan Credit | Wine Quality  | LFW    |
> > | :---- |  :----                  | :----        | :----        | :----        | :---- |
> > |$A$     | RedSoil (vsgrey/dampgrey) | FaultOther (y/n)   | Sex (male/female)         | isWhite (y/n)    | Sex (male/female)    |

---

> > > ### Author Response · Authors · 2021-11-23
> > > **Response to Reviewer wgFH (3/3)**
> > >
> > > **Q: It would have made it stronger if the authors mentioned how they would extend this work to non-binary sensitive attributes.**
> > >
> > > To handle the case where the sensitive feature is non-binary, we provide a **Generalized unfairness measure** definition, which replaces the penalty term $\mathbb U$ by the maximum over all the pairwise absolute difference of conditional reconstruction error between two groups, that is,
> > >     \\[
> > >         \mathbb U_{\max}(V, \mathbb{Q}) \triangleq \max_{(a, a') \in \mathcal A \times \mathcal A} | \mathbb E_{\mathbb{Q}}[\ell(V, X) | A = a] - \mathbb E_{\mathbb{Q}}[\ell(V, X) | A = a'] |.
> > >     \\]
> > >
> > > Moreover, we have generalized our reformulation result to this extended setting, which is stated below for convenience. The generalized reformulation result for the non-binary case allows us to apply a similar algorithm to the resulting optimization problem and retain the same convergence rate.
> > >
> > > **Theorem B.2** (Reformulation of non-binary fairness-aware PCA).
> > > Suppose that for any $a\in \mathcal A$, either of the following two conditions holds:
> > > 1) $0\le \lambda \le \hat p_a$,
> > > 2) the empirical second moment matrix $\hat M_a = \frac{1}{N_a} \sum_{i \in \mathcal I_a} \hat x_i \hat x_i^\top $ satisfies $\sum_{j = 1}^{d-k} \sigma_j(\hat M_a) \ge \varepsilon_a$, where $\sigma_j(\hat M_a)$ is the $j$-th smallest eigenvalues of $\hat M_a$.
> > > Then problem fairness-aware distributionally robust PCA with non-binary sensitive feature is equivalent to
> > > \begin{equation*}
> > >     \min\limits_{V \in \mathbb{R}^{d \times k}, V^\top V = I_k} \max_{a\neq a'} \left\lbrace \sum_{b \in \mathcal A} 2 c_{a,a',b} \sqrt{\varepsilon_b \langle I_d - VV^\top , \hat M_b \rangle} + \lambda \langle I_d - VV^\top , \hat M_a - \hat M_{a'} \rangle + \lambda (\varepsilon_a - \varepsilon_{a'}) \right\rbrace,
> > > \end{equation*}
> > > where the parameter $c_{a, a', b}$ admits values
> > > $$  c_{a,a',b} = \begin{cases}       \hat p_a + \lambda & \text{if } b = a,\\\       |\hat p_{a'} -\lambda| & \text{if } b = a', \\\      \hat p_b & \text{otherwise.}    \end{cases}$$
> > >
> > > Please see Appendix B in the revised manuscript for the detailed proof. Using the above Theorem, we can reparametrize the resulting optimization problem and apply a similar Riemannian geodesic descent algorithm to find the PCA. The above extension highlights the flexibility and power of our framework.
> > >
> > >
> > >
> > > **Q: On a very high level, it's not as obvious why reconstruction error should be balanced compared to, for example, in supervised learning. The toy example in Figure 1 is helpful but I'm wondering if the authors can describe a real-world example where bias in reconstruction error will be harmful - where PCA is the end result rather than a preprocessing step for supervised learning problem.**
> > >
> > > PCA can be equivalently interpreted by maximizing the variance of projected data, then the reconstruction error of a group is a proxy to determine how much its variances are retained. Hence, balancing subgroups’ reconstruction error has the potential to retain the importance of groups' variabilities.
> > >
> > > We agree with the reviewer that our paper does not provide any concrete answer on how and when a fair PCA may influence the fairness in the downstream tasks. The difficulty in answering  this question stems from the fact that PCA is too widely used, and there are a plethora of downstream tasks that can be implemented (classification, regression, object detection, etc.). This lack of understanding is also pertinent to any concurrent paper in fair PCA, including Samadi et al. (2018).

---

### Official Review · Reviewer_Hi3C · 2021-11-02

**Correctness:** 4
**Technical Novelty And Significance:** 2
**Empirical Novelty And Significance:** 2
**Recommendation:** 6
**Confidence:** 4

**Main Review:**

Strength:

1. The paper is nicely written and easy to follow. The motivation is clear and the problem formulation is novel.

2. Results on reformulation and riemannian gradient descent render tractable algorithms for solving the proposed problem. I do not find major technical errors.

3. Numerical experiments are based on real data set and are comprehensive.

Weakness:

1. The formulation simply combines distributioanlly robust PCA with a fairness penalty. Although this formulation is new, it seems a natural extension of existing frameworks.

2. The theoretical results are relatively easy to derive based on existing results on distributionally robust optimization and riemannian optimization. Hence, the originality is not methodological, but just applying well-established tools to a new formulation.

3. The dimensions of variables in UCI datasets are still relatively low (<=40 according to Table 2). The results can be strengthened by testing on datasets of larger dimensions.


**Summary Of The Paper:**

This paper studies the formulation, reformulation and algorithm for distributionally robust fairness-aware PCA. The reformulation exploits techniques from distributionally robust optimization, and the algorithm is based on reimannian sub-gradient descent. The theory is tested on UCI datasets.

**Summary Of The Review:**

This paper proposes a new formulation for PCA and derives a complete set of results with regard to the computational issues of the new results. My only reservation is that most theory are relatively straightforward compared to existing literature on distributionally robust optimization and Riemannian optimization.

---

> ### Author Response · Authors · 2021-11-22
> **Response to Reviewer Hi3C (1/1)**
>
> We thank the reviewer for the constructive comments and suggestions, we elaborated more on your concerns below, please have a look for more information
>
> **Q: The formulation simply combines distributionally robust PCA with a fairness penalty. Although this formulation is new, it seems a natural extension of existing frameworks.**
>
> Our paper blends ideas and techniques from two prominent fields of machine learning: (distributional) robustness and fairness. Nevertheless, we do not agree with the statement that our results are “natural extensions”. On the contrary, our results require many non-trivial designs which we elaborate below:
> - A separable ambiguity set: The ambiguity set $\mathbb B (\hat{\mathbb P})$ is constructed from individual ambiguity sets on the space of *conditional* probability distributions. Only by using this construct can we decompose the objective function into a pointwise maximum of two individual functions $J_0(V)$ and $J_1(V)$ which admit analytical reformulations.
> Note that in the existing literature of distributionally robust optimization, the ambiguity set is constructed on the *joint* distribution. Using a joint ambiguity set will not allow such decomposition.
> - Wasserstein-type distance: The Wasserstein-type distance is compulsory in order to obtain the reformulation in Proposition 3.3. If *any* other distance or divergence is used, then one may not be able to arrive at closed-form reformulations. We would like to emphasize that the closed-form expression of Proposition 3.3 is crucial for algorithmic analysis in Section 4. Thus, the choice of the Wasserstein-type distance has been decided with a specific purpose of facilitating the Riemannian geodesic descent algorithm.
> - Reparametrization: The reformulation with the variable $V$ is not convex, and we require an additional step to reparametrize the resulting problem in the complementary variable $U$ to obtain a convex problem. This reparametrization, despite being simple, is also crucial. Moreover, existing literature on optimization on the Stiefel manifold often makes a direct assumption on convexity (i.e., convex in $V$). Thus, on the modelling viewpoint, this complementary reparametrization is critical to resolve the convexity issue.
> - Condition verification: Our paper also verifies the conditions of convergence by providing an explicit formula for the Lipschitz constant in Lemma 4.2.
>
> With the above four points, we would like to emphasize that the results of this paper are non-trivial. In contrast, these results are the culmination of many appropriate modelling choices. If any of the above four points are removed or altered, the final results will no longer hold.
>
> **Q: The theoretical results are relatively easy to derive based on existing results on distributionally robust optimization and Riemannian optimization. Hence, the originality is not methodological, but just applying well-established tools to a new formulation.**
>
> Although our theoretical results rely on existing results from the literature of distributionally robust optimization and manifold optimization, they are not completely trivial but require some extra effort. For example, Proposition 3.3, which is the main tool we used in our reformulation results, is new and does not follow from any of the existing results in the DRO literature.
>
> For the proposed algorithm and its convergence rate, the novelty lies in reparametrizing the problem using the variable $U$, instead of the original variable $V$. Based on this reparameterization, we are able to apply the theoretical cited framework. Another contribution in this part is to establish an explicit upper bound on the Lipschitz constant in terms of the problem data, which is crucial to the application of the theoretical framework.
>
> **Q: The dimensions of variables in UCI datasets are still relatively low (<=40 according to Table 2). The results can be strengthened by testing on datasets of larger dimensions.**
>
> We include an additional experiment on a rescaled version of Labeled faces in the wild (LFW) datasets which has $576$ dimensions input. This LFW dataset is also the standard test dataset for fair PCA in Samadi et al. (2018).
>
> The table below shows the additional results for the LFW dataset:
>
> |         |  RFPCA |       |   FairPCA |       | CFPCA |  |
> | :---- | :----: | :----: | :----: | :----: | :----: | :----: |
> | Dataset | ABDiff. | ARE. | ABDiff. | ARE.  |ABDiff. &nbsp; ARE.  |
> |LFW-InSample | $\bf0.4147$ | $7.5137$ | $0.5300$ | $\bf7.5127$  |   fail to converge  |
> |LFW-OutofSample | $\bf 0.4463$ | $\bf 7.6229$ | $0.5340$ | $7.6361$|  fail to converge   |
>
> We can observe that our method outperforms Samadi et al. (2018) while the model from Olfat & Aswani (2019) fails to return any result (possibly due to high dimension). Details results and adjustments for LFW are included in the Appendix C and Appendix D.1

---

### Official Review · Reviewer_oUxn · 2021-11-03

**Correctness:** 3
**Technical Novelty And Significance:** 2
**Empirical Novelty And Significance:** 2
**Recommendation:** 5
**Confidence:** 5

**Main Review:**

Strength: + Robustness criterion for linear transformation of fairness can be formulated as a single optimization problem as long as the ambiguity set is defined using a single normal distribution.
+ A smooth reformulation is presented which can be solved using standard riemannian gradient descent algorithm.

Weakness: - Generalization to multiclass sensitive attribute, and other fairness metrics is missing.
- Figure 1 illustrating different approaches is nice, although it fails to capture the distributional robustness aspect of the formulation presented in the paper.
- It is hard to follow the paper since the notations are not standard.
- Experimental results are preliminary and conclusions are not clear.


**Summary Of The Paper:**

The paper presents an optimization based formulation of fair principal components analysis problem with a particular focus on the robustness of the optimal models. In the case when the desired robustness is specified by Gaussian distributions, the authors show how to adapt a manifold subgradient descent to solve a non-convex reformulation of the proposed fair pca formulation.


**Summary Of The Review:**

Justification:  The technical content is interesting, although in its current form, it is hard to follow. For example, why is problem (4)  infinite dimensional? As far as I can see, the decision variables in optimization problem (4) is a d by k matrix which is finite dimensional, although the zeroth order and first order oracles maybe noisy. This is evident in their convergence analysis, where they can only bound the gradient norm of the reformulated variable U (which in its original formulation is V). The paper contains some technical insights in the formulation as a SDP which happens to have a nice connection to Riemannian geometry as illustrated. However, at times, the paper reads as if one has to refer multiple textbooks just to make sure that paper indeed is technically correct. For example, many quantities appear without definitions (\hat{p}_a is not defined), and sometimes with somewhat unclear notations (Q(dx\times da). To me, the biggest technical question that the paper fails to answer is the generalization to nonbinary features. This is important in the context of the paper's contribution because we know that standard SDP solvers do not scale to settings with big or high dimensional data. The convergence rate in this setting is also not clear since a naive extension would indicate at least polynomial slowdown. The experiments are also inconclusive since the paper focuses exclusively on the binary sensitive attribute setting, and with just one metric. While the approach here may not be directly extendable to other metrics, the paper completely misses on discussing when the chosen formulation is appropriate for practitioners.

---

> ### Author Response · Authors · 2021-11-22
> **Response to Reviewer oUxn (1/2)**
>
> We thank the reviewer for the constructive comments. We have revised the manuscript thoroughly to take your comments into account. All the modifications to the manuscript are highlighted in blue.
> Please find below the detailed reply to your comments.
>
> **Q: Figure 1 illustrating different approaches is nice, although it fails to capture the distributional robustness aspect of the formulation presented in the paper.**
>
> The distributional robustness (in the form of a min-max optimization formulation) aims to improve the performance of the PCA when the available data is limited. This robustness is captured using the tuning parameter $\alpha$. In our numerical experiment, we typically observe that the optimal $\alpha$ obtained from cross-validation is non-zero. This implies that adding robustness improves the performance.
> The benefit of robustness can be demonstrated more easily using a toy example with two dimensions. We use two $2$-dimensional Gaussian distributions to represent two groups of a sensitive attribute, two distributions both have the mean at $(0,0)$ and covariance matrices are \\[\begin{pmatrix}  4.0 & 0 \\\  0 & 0.2 \end{pmatrix} \quad \text{and} \quad \begin{pmatrix}  0.2 & 0.4 \\\  0.4 & 3.0 \end{pmatrix},\\]  respectively. For each simulation, we independently generate $200$ samples for the majority group and $100$ samples for the minority group. The projections are learned on training data and measured on a fixed test dataset consisting of $8000$ samples from the majority, and $4000$ samples from the minority group. We measure the weighted summation of $\text{ARE.}$~and $\text{ABDiff.}$ with $\lambda=0.1$, and the result is averaged over $100$ simulations. Suppose that group $1$ is the minority group, we use $\varepsilon_1 = \alpha$ in which, $\alpha$ is tested with $100$ values evenly spaced in $[0, 1]$.
> The average loss collected from the experiment indicates that a value $\alpha > 0$ also minimizes the weighted error from the test dataset. This observation once again strengthens the needs for distributional robustness in the formulation. The detailed experiments are included in Appendix D.
>
> **Q: why is problem (4) infinite dimensional?**
>
> The infinite dimensionality comes from the inner maximization problem where the decision variable is a probability distribution residing in a certain subset of the infinite-dimensional space of probability distributions.
>
>
> **Q: Generalization to nonbinary features.**
>
> To handle the case where the sensitive feature is non-binary, we provide **Generalized unfairness measure** definition, which replaces the penalty term $\mathbb U$ by the maximum over all the pairwise absolute difference of conditional reconstruction error between two groups, that is,
>     \\[
>         \mathbb U_{\max}(V, \mathbb{Q}) \triangleq \max_{(a, a') \in \mathcal A \times \mathcal A} | \mathbb E_{\mathbb{Q}}[\ell(V, X) | A = a] - \mathbb E_{\mathbb{Q}}[\ell(V, X) | A = a'] |.
>     \\]
>
> Moreover, we have generalized our reformulation result to this extended setting, which is stated below for convenience. The generalized reformulation result for the non-binary case allows us to apply a similar algorithm to the resulting optimization problem and retain the same convergence rate.
>
> **Theorem B.2** (Reformulation of non-binary fairness-aware PCA).
> Suppose that for any $a\in \mathcal A$, either of the following two conditions holds:
> 1) $0\le \lambda \le \hat p_a$,
> 2) the empirical second moment matrix $\hat M_a = \frac{1}{N_a} \sum_{i \in \mathcal I_a} \hat x_i \hat x_i^\top $ satisfies $\sum_{j = 1}^{d-k} \sigma_j(\hat M_a) \ge \varepsilon_a$, where $\sigma_j(\hat M_a)$ is the $j$-th smallest eigenvalues of $\hat M_a$.
> Then problem fairness-aware distributionally robust PCA with non-binary sensitive feature is equivalent to
> \begin{equation*}
>     \min\limits_{V \in \mathbb{R}^{d \times k}, V^\top V = I_k} \max_{a\neq a'} \left\lbrace \sum_{b \in \mathcal A} 2 c_{a,a',b} \sqrt{\varepsilon_b \langle I_d - VV^\top , \hat M_b \rangle} + \lambda \langle I_d - VV^\top , \hat M_a - \hat M_{a'} \rangle + \lambda (\varepsilon_a - \varepsilon_{a'}) \right\rbrace,
> \end{equation*}
> where the parameter $c_{a, a', b}$ admits values
> $$  c_{a,a',b} = \begin{cases}       \hat p_a + \lambda & \text{if } b = a,\\\       |\hat p_{a'} -\lambda| & \text{if } b = a', \\\      \hat p_b & \text{otherwise.}    \end{cases}$$
>
> See Appendix B in the revised manuscript for the detailed proof.

---

> > ### Author Response · Authors · 2021-11-22
> > **Response to Reviewer oUxn (2/2)**
> >
> > **Q: The experiments are also inconclusive since the paper focuses exclusively on the binary sensitive attribute setting, and with just one metric.**
> >
> > To our knowledge, doing experiments on datasets with binary sensitive attributes is considered by a majority of previous work in the fairness community (e.g. Feldman et al. (2015); Hardt et al. (2016); Chouldechova (2017)) . Furthermore, it is sufficient to demonstrate the efficiency of fairness criteria with the binary sensitive attribute, which “is of central importance in many applications, encompassing the main conceptual and technical challenges” (Hardt et al. (2016), section 2).
> >
> > Regarding other metrics: We tested the methods on the fairness criterion from Olfat & Aswani (2019), which is based on a family of classifiers and bounds the classifier’s accuracy in predicting a sensitive attribute with inputs that are features from the PCA model. The results are presented in Table 4 in Appendix D.1. It is important to note that our method and Samadi et al. (2018) do not possess information about the classifier, and have a totally different goal from Olfat & Aswani (2019). It is thus natural that Olfat & Aswani (2019) dominate in this comparison.
> >
> > **Q: the paper completely misses on discussing when the chosen formulation is appropriate for practitioners.**
> >
> > Our formulation uses an unfairness penalty to regularize the objective function, and it is fundamentally different from existing approaches for fair PCA including Samadi et al. (2018) and Olfat & Aswani (2019). It is important to note that using an unfairness penalty is already examined in the settings of fair classification (e.g. Taskesen et al. (2020)) or fair regression (e.g. Scutari et al. (2021)), and they have been shown to deliver favorable empirical results.
> >
> > Using an unfairness penalty is also attractive because it offers flexibility in controlling the fairness level through the choice of the parameter $\lambda$. This is particularly suitable for the case where practitioners have prior knowledge about the problem and want to have control over the error - balance tradeoff. The practitioners can obtain the information about this trade-off by looking at the Pareto plot as described in Figure 2, and from this plot they can choose the desired balance.
> >
> > **References**
> >
> > Feldman, M., Friedler, S. A., Moeller, J., Scheidegger, C., & Venkatasubramanian, S. (2015, August). Certifying and removing disparate impact. In proceedings of the 21th ACM SIGKDD international conference on knowledge discovery and data mining (pp. 259-268).
> >
> > Hardt, M., Price, E., & Srebro, N. (2016). Equality of opportunity in supervised learning. Advances in neural information processing systems, 29, 3315-3323.
> >
> > Chouldechova, A. (2017). Fair prediction with disparate impact: A study of bias in recidivism prediction instruments. Big data, 5(2), 153-163.
> >
> > Taskesen, B., Nguyen, V. A., Kuhn, D., & Blanchet, J. (2020). A distributionally robust approach to fair classification. arXiv preprint arXiv:2007.09530.
> >
> > Scutari, M., Panero, F., & Proissl, M. (2021). Achieving Fairness with a Simple Ridge Penalty. arXiv preprint arXiv:2105.13817.

---

### Decision · Program_Chairs · 2022-01-20

**Decision:**

Accept (Poster)

**Comment:**

This paper considers the problem of distributionally robust fair PCA for binary sensitive variables. The main modeling contribution of the paper is the consideration of fairness and robustness of the PCA simultaneously, and the main technical contribution of the paper is the provision of a Riemannian subgradient descent algorithm for this problem and proof that it reaches local optima of this non-convex optimization problem. The results will be of interest to those working at the intersection of fair and robust learning.